# DeepMIP: Model intercomparison of early Eocene climatic optimum (EECO) large-scale climate features and comparison with proxy data

Daniel J. Lunt[1], Fran Bragg[1], Wing-Le Chan[2], David K. Hutchinson[3], Jean-Baptiste Ladant[4], Polina Morozova[5], Igor Niezgodzki[6,7], Sebastian Steinig[1], Zhongshi Zhang[8,9], Jiang Zhu[4], Ayako Abe-Ouchi[2], Eleni Anagnostou[10], Agatha M. de Boer[3], Helen K. Coxall[3], Yannick Donnadieu[11], Gavin Foster[12], Gordon N. Inglis[12], Gregor Knorr[6], Petra M. Langebroek[8], Caroline H. Lear[13], Gerrit Lohmann[6], Christopher J. Poulsen[4], Pierre Sepulchre[14], Jessica E. Tierney[15], Paul J. Valdes[1], Evgeny M. Volodin[16], Tom Dunkley Jones[17], Christopher J. Hollis[18], Matthew Huber[19], and Bette L. Otto-Bliesner[20]

[1]School of Geographical Sciences, University of Bristol, UK
[2]Atmosphere and Ocean Research Institute, University of Tokyo, Japan
[3]Department of Geological Sciences, Stockholm University, Sweden
[4]Department of Earth and Environmental Science, University of Michigan, USA
[5]Institute of Geography, Russian Academy of Sciences, Russia
[6]Alfred Wegener Institute, Helmholtz Centre for Polar and Marine Research, Bussestrasse 24, D-27570 Bremerhaven, Germany
[7]ING PAN - Institute of Geological Sciences Polish Academy of Sciences, Research Center in Kraków, Biogeosystem Modelling Group, Senacka 1, 31-002, Kraków, Poland
[8]NORCE Norwegian Research Centre, Bjerknes Centre for Climate Research, Norway
[9]China University of Geoscience (Wuhan), China
[10]GEOMAR Helmholtz-Zentrum für Ozeanforschung Kiel, Germany
[11]CNRS, France
[12]School of Ocean and Earth Science, National Oceanography Centre Southampton, University of Southampton, UK.
[13]School of Earth and Environmental Sciences, Cardiff University, UK
[14]Laboratoire des Sciences du Climat et de l'Environnement, LSCE/IPSL, CEA-CNRS-UVSQ, Université Paris-Saclay, France
[15]Department of Geosciences, University of Arizona, USA
[16]Institute of Numerical Mathematics, Moscow, Russia
[17]School of Geography, Earth and Environmental Sciences, Birmingham University, UK
[18]GNS Science, New Zealand
[19]Department of Earth, Atmospheric, and Planetary Sciences, Purdue University, West Lafayette, Indiana, USA
[20]Climate and Global Dynamics Laboratory, National Center for Atmospheric Research, Boulder, 80305, USA

**Correspondence:** Dan Lunt (d.j.lunt@bristol.ac.uk)

**Abstract.** We present results from an ensemble of eight climate models, each of which has carried out simulations of the early Eocene climate optimum (EECO, ∼50 million years ago). These simulations have been carried out in the framework of DeepMIP (www.deepmip.org), and as such all models have been configured with the same paleogeographic and vegetation boundary conditions. The results indicate that these non-$CO_2$ boundary conditions contribute between 3 and 5°C to Eocene warmth. Compared to results from previous studies, the DeepMIP simulations show in general reduced spread of global mean

surface temperature response across the ensemble for a given atmospheric $CO_2$ concentration, and an increased climate sensitivity on average. An energy balance analysis of the model ensemble indicates that global mean warming in the Eocene compared with preindustrial arises mostly from decreases in emissivity due to the elevated $CO_2$ (and associated water vapour and long-wave cloud feedbacks), whereas in terms of the meridional temperature gradient, the reduction in the Eocene is

primarily due to emissivity and albedo changes due to the non-$CO_2$ boundary conditions (i.e. removal of the Antarctic ice sheet and changes in vegetation). Three of the models (CESM, GFDL, and NorESM) show results that are consistent with the proxies in terms of global mean temperature, meridional SST gradient, and $CO_2$, without prescribing changes to model parameters. In addition, many of the models agree well with the first-order spatial patterns in the SST proxies. However, at a more regional scale the models lack skill. In particular, in the southwest Pacific, the modelled anomalies are substantially less

than indicated by the proxies; here, modelled continental surface air temperature anomalies are more consistent with surface air temperature proxies, implying a possible inconsistency between marine and terrestrial temperatures in either the proxies or models in this region. Our aim is that the documentation of the large scale features and model-data comparison presented herein will pave the way to further studies that explore aspects of the model simulations in more detail, for example the ocean circulation, hydrological cycle, and modes of variability; and encourage sensitivity studies to aspects such as paleogeography,

orbital configuration, and aerosols.

## 1  Introduction

Paleoclimate model-data comparisons allow us to assess confidence in the results from model sensitivity studies that explore the mechanisms that drove past climate change, and allow us to assess confidence in the future climate predictions from

these models. Past warm climates, particularly those associated with high atmospheric $CO_2$ concentrations, are especially relevant because they share characteristics with possible future climates (Burke et al., 2018). In this context, there has been a community focus on the Pliocene ($\sim$3-5 million years ago;  Haywood et al., 2013) and Eocene ($\sim$50 million years ago;  Lunt et al., 2012), which provide natural examples of past worlds with high $CO_2$ concentrations of $\sim$300-400 ppmv and $\sim$1200-2500 ppmv respectively. In this paper, we focus on the Eocene, presenting model results that have recently been carried out in

the framework of the DeepMIP project (www.deepmip.org; Lunt et al., 2017; Hollis et al., 2019), and associated model-data comparisons. Given the similarity of Eocene $CO_2$ concentrations and climate to those that are attained under high growth/low mitigation future scenarios considered by the IPCC (Burke et al., 2018), the Eocene provides a potential test-bed for state-of-the-art climate model predictions of the future.

Eocene modelling and model-data comparisons have a long history (e.g. Barron, 1987; Sloan and Barron, 1992). More

recently, Lunt et al. (2012) carried out a synthesis of a group of models that had all carried out Eocene simulations (Lunt et al., 2010b; Heinemann et al., 2009; Winguth et al., 2010; Huber and Caballero, 2011; Roberts et al., 2009), with a focus on surface

temperatures. Subsequent work also explored the precipitation in the simulations (Carmichael et al., 2016) and the implications for ice sheet growth (Gasson et al., 2014). This was an "ensemble of opportunity" in that the model simulations were carried out independently, using a variety of paleogeographical and vegetation boundary conditions, and carried out under a range of

40 different $CO_2$ concentrations. A proxy data synthesis was also produced as part of the Lunt et al. (2012) study, consisting of sea surface temperatures (SSTs), and a previously compiled continental temperature dataset (Huber and Caballero, 2011). That model-data comparison showed that: (a) For a given $CO_2$ concentration, there was a wide spread in global mean temperature response across the models. For example, at $CO_2$ concentrations $\times 4$ those of preindustrial, the range in modelled global mean continental near-surface air temperature was 5.8°C. (b) Given $CO_2$ concentrations of $\times 16$, the CCSM3 model was able

to reproduce the mean climate and meridional temperature gradient indicated by the proxies. (c) The HadCM3 model had relatively weak polar amplification compared with the other models. (d) The climate sensitivity across the models was fairly similar, but HadCM3 had a notable non-linearity in sensitivity, in contrast to CCSM3. (e) Interpreting middle and high latitude proxy SSTs as representing summer temperatures brought the modelled temperatures closer to those indicated by the proxies.

At that time, due to uncertainties in pre-ice core $CO_2$ proxies, it was not possible to rule out the high $CO_2$ concentrations

needed by CCSM3 to match the proxies, although such high values were outside the range of many $CO_2$ compilations (Beerling and Royer, 2011). As such, the Intergovernmental Panel on Climate Change (IPCC) concluded that "While recent simulations of the EECO... exhibit a wide inter-model variability, there is generally good agreement between new simulations and data, particularly if seasonal biases in some of the marine SST proxies from high-latitude sites are considered" (Masson-Delmotte et al., 2013). However, more recent work has indicated that early Eocene $CO_2$ concentration were in the range 1170 ppmv

to 2490 ppmv (95% confidence interval) (Anagnostou et al., 2020), substantially lower than the $\times 16$ (4480 ppmv) CCSM3 simulation that was the best fit to proxy data of the models examined in Lunt et al. (2012).

Following on from that initial modelling work, two studies (Sagoo et al., 2013; Kiehl and Shields, 2013) showed that the representation of clouds in models could be modified to give greater polar amplification and climate sensitivity, resulting in simulations that were more consistent with temperature proxies of the Eocene at lower $CO_2$. Kiehl and Shields (2013)

decreased the cloud drop density and increased the cloud drop radius to represent the effect of reduced cloud-condensation nucleii in the Eocene compared with modern, and found that, at a $CO_2$ concentration of 1375 ppmv and $CH_4$ of 760 ppbv (their "pre-PETM" simulation) they obtained a good agreement with data. Sagoo et al. (2013) perturbed ten atmospheric and oceanic variables in an ensemble, of which those associated with clouds were judged the most important, and found that two ensemble members were able to simulate temperatures in good agreement with proxies at a $CO_2$ concentration of 560 ppmv.

Although both of these studies indicated that clouds could be the key to reconciling proxies and models, neither of the changes applied were physically based. Furthermore, more recent work has indicated that the response to modifying cloud albedo is very similar to that of increasing $CO_2$, at least in terms of meridional temperature gradient (Carlson and Caballero, 2017), such that prescribing cloud changes can result in a system that is somewhat unconstrained. As such, the relevance of these studies for future prediction or to other paleo time periods remains unclear.

To facilitate an intermodel comparison, a standard set of boundary conditions and experimental design has been proposed for a coordinated set of model simulations of the early Eocene (Lunt et al., 2017). In addition, there has been a community effort to

better characterise the uncertainties in proxy temperature and $CO_2$ estimates of the latest Paleocene, Paleocene–Eocene Thermal Maximum (PETM) and EECO (Hollis et al., 2019). Furthermore, some models are available for deep-time paleoclimate simulations that are more advanced than those used in the Lunt et al. (2012) study; for example CESM1.2 includes a more advanced cloud microphysics scheme compared with CCSM3, HadCM3 has a higher ocean resolution than HadCM3L, and INMCM is a CMIP6-class model and therefore can be considered state-of-the-art. In this paper, we present an ensemble of early Eocene simulations from a range of climate models, carried out in this framework, and compare them with the latest paleo data of the EECO. Three key scientific questions that we address are:

- What are the large-scale features of the DeepMIP Eocene simulations?

- What are the causes of the model spread in these simulations?

- How well do the models fit the proxy data, and has there been an improvement in model fit compared with previous work?

## 2 DeepMIP model simulations

Here we briefly describe the standard experimental design, and for each model give a brief description of the model and any departures from the standard experimental design.

### 2.1 Experimental design

The standard experimental design for the DeepMIP model simulations, and underlying motivation, is described in detail in Lunt et al. (2017). In brief, the simulations consist of a preindustrial control, and a number of Eocene simulations at various atmospheric $CO_2$ concentrations ($\times 3$, $\times 6$, and $\times 12$, although in practice many groups chose different concentrations; see Table 1). The paleogeography, vegetation, and river routing for the Eocene simulations are prescribed according to the reconstructions of Herold et al. (2014) (see Figure 3a,b and 4 in Lunt et al., 2017). The solar constant, orbital configuration, and non-$CO_2$ greenhouse gas concentrations are set to preindustrial values. Soil properties are set to homogeneous global mean values derived from the preindustrial simulation, and there are no continental ice sheets in the Eocene simulations. A suggested initial condition for ocean temperature and salinity was given, but many groups diverged from this. The prescription of calculation of atmospheric aerosols were left to each individual group's discretion.

### 2.2 Individual model simulations

An overview of the model simulations is presented in Table 1 and in Table S1. Here we describe each model in turn, and the experimental design of the simulations where this diverged from that described in Lunt et al. (2017).

**Table 1.** Summary of the DeepMIP Eocene model simulations described and presented in this paper. In addition to the simulations listed, each model has an associated preindustrial control. More information about the spinup of each simulation is in the Supplement, Table S2. In this paper, each model is referred to by its Short Name. For GFDL and IPSL, the reference describes a related, but not identical, set of paleo simulations to those described in this paper.

| Model | Short Name | $CO_2$ | CMIP generation | Simulation reference |
|---|---|---|---|---|
| CESM1.2_CAM5 | CESM | $\times 1, \times 3, \times 6, \times 9$ | CMIP5 | Zhu et al. (2019) |
| COSMOS-landveg_r2413 | COSMOS | $\times 1, \times 3, \times 4$ | CMIP3 | *This paper* |
| GFDL_CM2.1 | GFDL | $\times 1, \times 2, \times 3, \times 4, \times 6$ | CMIP3 | *This paper* |
| HadCM3B_M2.1aN | HadCM3 | $\times 1, \times 2, \times 3$ | CMIP3 | *This paper.* |
| INM-CM4-8 | INMCM | $\times 6$ | CMIP6 | *This paper* |
| IPSLCM5A2 | IPSL | $\times 1.5, \times 3$ | CMIP5 | Zhang et al. (2020) |
| MIROC4m | MIROC | $\times 3$ | CMIP3 | *This paper* |
| NorESM1_F | NorESM | $\times 2, \times 4$ | CMIP5-6 | *This paper* |

### 2.2.1 CESM (CESM1.2_CAM5)

*CESM model description*

The Community Earth System Model version 1.2 (CESM) is used, which consists of the Community Atmosphere Model 5.3 (CAM), the Community Land Model 4.0 (CLM), the Parallel Ocean Program 2 (POP), the Los Alamos sea ice model 4 (CICE), the River Transport Model (RTM), and a coupler connecting them (Hurrell et al., 2013). In comparison to previous versions of the CESM models that have been used for Eocene simulation, e.g. CCSM3 (Huber and Caballero, 2011; Winguth et al., 2010; Kiehl and Shields, 2013) and CESM1(CAM4) (Cramwinckel et al., 2018), CESM1.2(CAM5) represents a nearly complete overhaul of physical parameterizations in the atmosphere model, including new schemes for radiation, boundary layer, shallow convection, cloud microphysics and macrophysics, and aerosols (Hurrell et al., 2013). The new two-moment microphysical scheme predicts both the cloud water mixing ratio and particle number concentration. The new aerosol scheme predicts the aerosol mass and number, and is coupled with the cloud microphysics, allowing the inclusion of aerosol indirect effects. The new boundary layer and shallow convection schemes improve the simulation of shallow clouds in the marine boundary layer. These new parameterizations in CAM5 produce a cloud simulation that agrees much better with satellite observations (Kay et al., 2012) and a larger present-day equilibrium climate sensitivity ($\sim 4°C$) than previous versions ($\sim 3°C$) (Gettelman et al., 2012). CESM1.2(CAM5) reproduces key features in state and variability of past climates, including the mid-Piacenzian Warm period (Feng et al., 2019), the Last Glacial Maximum (Zhu et al., 2017a), Heinrich events (Zhu et al., 2017b), and the last millennium (Otto-Bliesner et al., 2015; Thibodeau et al., 2018). To make the model suitable for a paleoclimate simulation with a high $CO_2$ level, the model code has been slightly modified to incorporate an upgrade to the radiation code that corrects

the missing diffusivity angle specifications for certain longwave bands. As a result of the code modification, CAM5 has been re-tuned with a different relative humidity threshold for low clouds (rhminl=0.8975, versus the default value of 0.8875). These code and parameter changes are not found to alter the present-day climate sensitivity in CESM (Zhu et al., 2019).

*CESM model simulations*

The CESM Eocene simulations are run at $\times 1$, $\times 3$, $\times 6$, and $\times 9$ $CO_2$ concentrations (Table 1). The atmosphere and land have a horizontal resolution of $1.9 \times 2.5°$ (latitude $\times$ longitude) with 30 hybrid sigma-pressure levels in the atmosphere. The ocean and sea ice are on a nominal $1°$ displaced pole Greenland grid with 60 vertical levels in the ocean. CAM5 runs with a prognostic aerosol scheme with prescribed preindustrial natural emissions that have been redistributed according to the Eocene

paleogeography following the method in Heavens et al. (2012). The vegetation type from Herold et al. (2014) is prescribed in the land model with active carbon and nitrogen cycling. A modified marginal sea balancing scheme was applied for the Arctic Ocean, which removes any gain/deficit of freshwater over the Arctic Ocean and redistributes the mass evenly over the global ocean surface excluding the Arctic. This implementation conserves ocean salinity and is necessary to prevent the occurrence of negative salinity that results from high precipitation and river runoff under warm conditions. A similar balancing scheme has

been included for marginal seas in all the previously published CESM simulations (Smith et al., 2010). The ocean temperature and salinity were initialized from a previous PETM simulation using CCSM3 (Kiehl and Shields, 2013). The sea ice model was initialized from a sea ice-free condition. All simulations have been integrated for 2000 model years, with the exception of $\times 1$ which was run for 2600 model years.

### 2.2.2 COSMOS (COSMOS-landveg_r2413)

*COSMOS model description*

The atmosphere is represented by means of the ECHAM5 atmosphere general circulation model (Roeckner et al., 2003). ECHAM5 is based on a spectral dynamical core and includes 19 vertical hybrid sigma-pressure levels. The series of spectral harmonics is curtailed via triangular truncation at wave number 31 (approx. $3.75° \times 3.75°$). Ocean circulation and sea ice dynamics are computed by the MPIOM ocean general circulation model (Marsland et al., 2003) that is employed at 40

unequally-spaced levels on a bipolar curvilinear model grid with formal resolution of $3.0° \times 1.8°$ longitude by latitude. The coupled model ECHAM5/MPIOM is described by Jungclaus et al. (2006). A concise description of the application of the COSMOS for paleoclimate studies is given by Stepanek and Lohmann (2012). The COSMOS version used here has proven to be a suitable tool for the study of the Earth's past climate, from the Holocene (Wei and Lohmann, 2012; Wei et al., 2012; Lohmann et al., 2013) and previous interglacials (Pfeiffer and Lohmann, 2016; Gierz et al., 2017), glacial (Gong et al., 2013;

Zhang et al., 2013, 2014; Abelmann et al., 2015; Zhang et al., 2017) to tectonic time scales (Knorr et al., 2011; Knorr and Lohmann, 2014; Walliser et al., 2016; Huang et al., 2017; Niezgodzki et al., 2017; Stärz et al., 2017; Walliser et al., 2017; Vahlenkamp et al., 2018; Niezgodzki et al., 2019). The standard model code of the Community Earth System Models (COSMOS) version COSMOS-landveg r2413 (2009) is available upon request from the Max Planck Institute for Meteorology in Hamburg (https://www.mpimet.mpg.de).

*COSMOS model simulations*

The COSMOS simulations are carried out at $\times 1$, $\times 3$, and $\times 4$ preindustrial $CO_2$ concentrations of 280 ppm. The ocean temperatures in $3\times$ $CO_2$ concentration were initialized with uniformly horizontal and vertical temperatures of $10°$ Celsius. The initial ocean salinity was set to 34.7 psu. The simulations with $1\times$ and $4\times$ $CO_2$ concentrations were restarted from $3\times$ $CO_2$ after 1000 years. All simulations were run with transient orbital configurations until year 8000. Subsequently, they were run for 1500 years (to the year 9500), with fixed, preindustrial orbital parameters. All simulations employ the hydrological discharge model of Hagemann and Dümenil (1998) instead of the river routing provided by Herold et al. (2014).

### 2.2.3   GFDL (GFDL_CM2.1)

*GFDL model description*

These simulations use a modified version of GFDL CM2.1 (Delworth et al., 2006), similar to the late Eocene configuration in Hutchinson et al. (2018, 2019). The ocean component uses the modular ocean model (MOM) version 5.1.0, while the other components of the model are the same as in CM2.1; Atmosphere Model 2, Land Model 2 and the Sea Ice Simulator 1. The ocean and sea ice components use a horizontal resolution of $1°$ latitude $\times$ $1.5°$ longitude. A tripolar grid is used as in Hutchinson et al. (2018), with a regular latitude-longitude grid south of $65°$ N, and a transition to a bipolar Arctic grid north of $65°$ N, with poles over North America and Eurasia. There is no refinement of the latitudinal grid spacing in the tropics. The ocean uses 50 vertical levels, with the same vertical spacing as CM2.1 The atmopsheric horizontal grid resolution is $3° \times 3.75°$, with 24 vertical levels, as in CM2Mc (Galbraith et al., 2010). This configuration enables relatively high resolution ocean and coastlines, with the advantage of a faster-running atmosphere. The topography (both land and ocean) uses the 55 Ma reconstruction of Herold et al. (2014), re-gridded to the ocean and atmosphere components. Manual adjustments are made to ensure that no isolated lakes or seas exist, and that any narrow ocean straits are at least 2 grid cells wide to ensure non-zero velocity fields. The minimum depth of ocean grid cells is 25 m; any shallow ocean grid cells are deepened to this minimum depth. In the atmosphere, the topography is smoothed using a 3-point mean filter to ensure a smoother interaction with the wind field. This was introduced to remove numerical noise over the Antarctic continent, due to convergence of meridians on the topography grid. Vegetation types are based on Herold et al. (2014), adapted to the corresponding vegetation type in CM2.1. Aerosol forcing is also adapted from Herold et al. (2014) to the model, and is a fixed boundary condition. Ocean vertical mixing is identical to that in Hutchinson et al. (2018); i.e. a uniform bottom-roughness enhanced mixing with a background diffusivity of $1.0 \times 10^{-5}$ m$^2$ s$^{-1}$.

*GFDL model simulations*

The model was initiated from idealised conditions, similar to those outlined in Lunt et al. (2017) with reduced initial temperatures: T($°$C) = (5000-z)/5000 * 25 $\cos(\phi)$ + 10 if z $\leq$ 5000 m and T($°$C) = 10 if z > 5000 m; where $\phi$ is latitude, $z$ is the depth of the ocean (positive downwards). The initial salinity was a constant of 34.7 psu. The above initial conditions were used for the $\times 1$, $\times 2$, $\times 3$ and $\times 4$ $CO_2$ experiments. These simulations were initially run for 1500 years, after which the ocean temperatures were adjusted in order to accelerate the approach to equilibrium. This adjustment consisted of calculating the average temperature trend for the last 100 years at each model level below 500m, taking a level-by-level global average of this trend, and applying a 1000-year extrapolation uniformly across the ocean at that level. This choice was based on the observation that

all model levels below the mixed layer were consistently cooling at a slow rate, and the rate of temperature adjustment was consistent over a long time scale. After a further 500 years, a second adjustment using the same method was performed. After the second adjustment, all simulations were continuously integrated with no further adjustments for a further 4000 years. Thus the simulations were run for a total of 6000 years. For the $\times 6$ $CO_2$ experiment, the initial conditions described above led to transient instabilities due to overheating the surface. So the $\times 6$ experiment was instead initialised using a globally uniform temperature of 19.32 °C. This represents the same global average temperature as in the other experiments, hence the same total ocean heat content. For the $\times 6$ $CO_2$, no step-wise adjustments were made; the model was run continuously for 6000 years.

#### 2.2.4 HadCM3 (HadCM3B_M2.1aN)

*HadCM3 model description*

The HadCM3 simulations are carried out with the HadCM3B-M2.1aN version of the model, as described in detail in Valdes et al. (2017). Equations are solved on a Cartesian grid with horizontal resolutions of $3.75° \times 2.5°$ in the atmosphere and $1.25° \times 1.25°$ in the ocean with 19 and 20 vertical levels, respectively. A few changes are made to the version described in Valdes et al. (2017) to make it suitable for deep-time paleoclimate modelling: (a) A salinity flux correction is applied to the global ocean (at all model depths) in order to conserve salinity. (b) The various modern-specific parameterisations in the ocean model are turned off, such as associated with Mediterranean and Hudson Bay outflow, and North Atlantic mixing. (c) A prognostic 1D ozone scheme is used instead of a fixed vertical profile of ozone. The standard configuration uses a prescribed ozone climatology which is a function of latitude, height, and month of the year that does not change with climate and can become numerically unstable at high $CO_2$ levels. The prognostic ozone scheme uses the diagnosed model tropopause height to assign three distinct ozone concentrations for the troposphere, tropopause, and stratosphere ($2.0x10^{-8}$, $2.0x10^{-7}$ and $5.5x10^{-6}$ in mmr, respectively). This allows for a dynamic update of the 1D ozone field in response to the thermally driven vertical expansion of the troposphere. Absolute values for the three levels are chosen to minimise the effects on global mean and overall tropospheric temperature changes compared to the standard 2D climatology. Concentrations at the uppermost model level are fixed to the higher stratospheric value to constrain the lower bound of total stratospheric ozone. Significant differences to the standard configuration are limited to the stratospheric meridional temperature gradient and zonal winds and are related to the missing latitudinal variations in the 1D field. Although HadCM3 has been used previously to simulate the Pliocene (e.g. Lunt et al., 2008, 2010a), the presented simulations represent the first published application of HadCM3 to pre-Pliocene boundary conditions. However, the lower resolution HadCM3L model has been previously used to simulate a range of pre-Quaternary climates (e.g. Lunt et al., 2016; Farnsworth et al., 2019a, b)

*HadCM3 model simulations*

The HadCM3 simulations are carried out at $\times 1$, $\times 2$, and $\times 3$ $CO_2$ concentrations. Several ocean gateways were artificially widened to allow unrestricted throughflow and maximum water depths in parts of the Arctic Ocean were reduced. The ocean temperatures were initialised from the final state of Eocene model simulations using HadCM3L. The HadCM3L simulations were set up identically to the corresponding HadCM3 simulations, but with lower ocean resolution ($3.75° \times 2.5°$ as opposed to $1.25° \times 1.25°$). The HadCM3L simulations were initialised from a similar idealised temperature and salinity state as described

in Lunt et al. (2017), but with a function that scales with $cos^2(lat)$ rather than $cos(lat)$ and overall reduced initial temperatures to ensure numerical stability in tropical regions. Ocean temperatures below 600 m were set to constant values of 4, 8 and 10 °C (at ×1, ×2, and ×3 $CO_2$ respectively) based on results from previous Ypresian simulations. The HadCM3 simulations were branched off from the respective HadCM3L integrations after 4400 to 4900 years of spin up and run for a further 2950 years. The initial 50 years of all HadCM3 runs used the simplified vertical diffusion scheme from HadCM3L (Valdes et al., 2017) to reduce numerical problems caused by the changed horizontal ocean resolution. The remaining years of the runs use the standard HadCM3 diffusion scheme (Valdes et al., 2017).

### 2.2.5 INMCM (INM-CM4-8)

*INMCM model description*

The INMCM simulations are carried out with the INM-CM48 (INM-CM4-8) version of the model, as described in Volodin et al. (2018). The INM-CM4-8 climate model has a horizontal resolution of $2° \times 1.5°$ in the atmosphere; 17 vertical sigma levels up to a value of 0.01 (about 30 km) are used for the Eocene experiment, and 21 levels for the preindustrial experiment. The equations of the atmosphere dynamics are solved by finite-difference methods. The parameterisations of physical processes correspond to the INM-CM5 model (Volodin et al., 2017). Parameterisation of condensation and cloud formation follows Tiedtke (1993). Cloud water is a prognostic variable. Parameterization of cloud fraction follows Smagorinsky (1963); cloud fraction is a diagnostic variable. The surface, soil and vegetation scheme follows Volodin and Lykossov (1998). The evolution of the equations for temperature, soil water and soil ice are solved at 23 levels from the surface to 10 meters depth. The fractional area of 13 types of potential vegetation is specified. Actual vegetation as well as LAI is calculated according to the soil water content in the root zone and soil temperature. This model also contains a carbon cycle and an aerosol scheme (Volodin and Kostrykin, 2016), taking into account the direct impact of aerosols on radiation, and the first indirect effect (the influence of aerosols on the condensation rate). The concentration of 10 types of aerosol and their radiative properties are calculated interactively. In the ocean component, the resolution of the INM-CM4-8 model is $1.0 \times 0.5$ degrees in longitude and latitude and has 40 sigma levels vertically. Finite difference equations are solved on a generalized spherical C-grid with the North Pole shifted to Siberia; the South Pole is at the same place as the geographical pole.

*INMCM model simulations*

The INM-CM4-8 Eocene simulation is carried out at ×6 $CO_2$ concentration. The INM-CM4-8 simulation was initialised from a similar idealised temperature and salinity state as described in Lunt et al. (2017), but the initial formula for the ocean temperature is modified: T=((5000-z)/5000 * 20 cos ($\phi$) ) +15 , reducing the initial temperatures into ensure numerical stability in tropical regions. The 27 biomes were converted into the 13 model types of vegetation. The duration for the Eocene simulation is 1150 years. Output data is averaged over years 1051-1150.

### 2.2.6 IPSL (IPSLCM5A2)

*IPSL model description*

The IPSL simulations are performed with the IPSL-CM5A2 earth system model (Sepulchre et al., 2020). IPSL-CM5A2 is

based on the CMIP5-generation previous IPSL earth system model IPSL-CM5A (Dufresne et al., 2013) but includes new revisions of each components, a re-tuning of global temperature, and technical improvements to increase computing efficiency. It consists of the LMDZ5 atmosphere model, the ORCHIDEE land surface and vegetation model and the NEMOv3.6 ocean model, which includes the LIM2 sea ice model and the PISCES-v2 biogeochemical model. LMDZ5 and ORCHIDEE run at a horizontal resolution of $1.9 \times 2.5°$ (latitude $\times$ longitude) with 39 hybrid sigma-pressure levels in the atmosphere. NEMO runs on a tripolar grid at a nominal resolution of $2°$, enhanced up to $0.5°$ at the Equator, with 31 vertical levels in the ocean. The performances and evaluation of IPSL-CM5A2 on preindustrial and historical climates are fully described in Sepulchre et al. (2020). Sepulchre et al. (2020) also provides a description of the technical changes that were implemented in IPSL-CM5A2 to carry out deep time paleoclimate simulations. In particular, the tripolar mesh grid on which NEMO runs has been modified to ensure that there are no singularity points within the ocean domain. Modern parameterizations of water outflows across specific straits, such as the Gibraltar or Red Sea straits, are also turned off.

*IPSL model simulations*

The IPSL simulations are run at $\times1.5$ and $\times3$ $CO_2$ concentrations. The bathymetry is obtained from the Herold et al. (2014) dataset, with additional handmade corrections in some locations, for instance in the West African region, to maintain sufficiently large oceanic straits. Modern boundary conditions of NEMO include forcings of the dissipation associated with internal wave energy for the M2 and K1 tidal components (de Lavergne et al., 2019). The parameterization follows Simmons et al. (2004) with refinements in the modern Indonesian Through Flow (ITF) region according to Koch-Larrouy et al. (2007). To create an Early Eocene tidal dissipation forcing, the Herold et al. (2014) M2 tidal field (obtained from the tidal model simulations of Green and Huber (2013)) is directly interpolated onto the NEMO grid using bilinear interpolation. In the absence of any estimation for the Early Eocene, the K1 tidal field is prescribed to 0. In addition, the parameterization of Koch-Larrouy et al. (2007) is not used here because the ITF does not exist in the Early Eocene. The geothermal heating distribution is created from the 55 Ma global crustal age distribution of Müller et al. (2008), on which is applied the age-heatflow relationship of the Stein and Stein (1992) model: $q(t) = 510 \times t^{-1/2}$ if $t \leq 55$ Ma and $q(t) = 48 + 96 \exp(-0.0278 \times t)$ if $t > 55$Ma. In regions of subducted seafloor where age information is not available, the minimal heatflow value is prescribed, derived from known crustal age. The $1° \times 1°$ resulting field is then bilinearly interpolated onto the NEMO grid. It must be noted that the Stein and Stein (1992) parameterization becomes singular for young crustal ages, which yields unrealistically large heatflow values. Following Emile-Geay and Madec (2009), an upper limit of 400 mW m$^{-2}$ is set for heatflow values after the interpolation procedure. Salinity is initialized as globally constant to a value of 34.7 psu following Lunt et al. (2017). The initialization of the model with the proposed DeepMIP temperature distribution (Lunt et al., 2017) led to severe instabilities of the model during the spin-up phase. The initial temperature distribution has thus been modified to follow: $T(°C) = (1000-z)/1000 * 25 \cos(\phi)$ + 10 if $z \leq 1000$ m and $T(°C) = 10$ if $z > 1000$ m. With $\phi$ the latitude and z the depth of the ocean (meters below surface). This new equation gives an initial globally constant temperature of $10°C$ below 1000 m and a zonally symmetric distribution above, reaching surface values of $35°C$ at the equator and $10°C$ at the poles. This corresponds to a $5°C$ surface temperature reduction compared to DeepMIP guidelines (Lunt et al., 2017). No sea ice is prescribed at the beginning of the simulations. In IPSL-CM5A2, the NEMO ocean model is inherently composed of the PISCES biogeochemical model. Biogeochemical cycles

and marine biology are directly forced by dynamical variables of the physical ocean and may affect the ocean physics via its influence on chlorophyll production, which modulates light penetration in the ocean. However, because this feedback does not much affect the ocean state (Kageyama et al., 2013) and because the early Eocene mean ocean colour is unknown, a constant chlorophyll value of 0.05 g.Chl/L is prescribed for the computation of light penetration in the ocean. As a consequence, marine biogeochemical cycles and biology do not alter the dynamics of the ocean and as such biogeochemical initial and boundary conditions have been kept to modern. The topographic field is created from the Herold et al. (2014) topographic dataset; LMDZ includes a subgrid scale orographic drag parameterization that requires high-resolution surface orography (Lott and Miller, 1997; Lott, 1999). A similar procedure is applied for the standard deviation of orography provided by Herold et al. (2014). Aerosol distributions are left identical to preindustrial values. The ×3 simulation is initialized from rest and run for 4000 years. The ×1.5 simulation is branched from year 1500 of the ×3 simulation and run for 4000 years. The ×1.5 and ×3 simulations are identical to those presented in Zhang et al. (2020).

### 2.2.7  MIROC (MIROC4m)

*MIROC model description*

The version of the Model for Interdisciplinary Research on Climate (MIROC) used here is MIROC4m, a mid-resolution model composed of atmosphere, land, river, sea ice and ocean components. Full documentation of the model can be found in K-1 model developers (2004) and a summary in Chan et al. (2011). The atmosphere has a horizontal resolution of T42 and 20 vertical sigma levels. Details of the land-surface model, Minimal Advanced Treatments of Surface Interaction and Runoff (MATSIRO), can be found in Takata et al. (2003). The ocean component is basically version 3.4 of the CCSR Ocean Component Model (COCO) - refer to Hasumi (2000). The horizontal resolution is set to 256×196, with higher resolution in the tropics, and the vertical resolution is set to 44 levels, with the top 8 in sigma coordinates. Present day bathymetry is derived from ETOPO5 data. For present day experiments, areas of water such as the Hudson Bay and the Mediterranean Sea are represented as isolated basins. As such, ocean salinity and heat are artificially exchanged with the open ocean through a 2-way linear damping. This damping and all isolated basins and lakes are removed in the DeepMIP simulation.

*MIROC model simulations*

Out of the three standard DeepMIP simulations, MIROC is used with ×3 $CO_2$ concentration only and run for 5000 model years. The atmosphere is initialised from a previous experiment without ice sheets and with ×2 $CO_2$ concentration. For the initial ocean state, salinity is set to a constant value of 34.7 psu, as recommended in Lunt et al. (2017). However, the ocean temperatures are 15°C cooler than those recommended, i.e. T(°C) = (5000-z)/5000 * 25 cos($\phi$) if z ≤ 5000 m and T(°C) = 0 if z > 5000 m. Previous MIROC experiments similar to this ×3 $CO_2$ DeepMIP simulation show that this initialisation should be much closer to the final climate state. Simulations were also carried out at ×1 and ×2 but they are not discussed in this paper.

### 2.2.8  NorESM (NorESM1_F)

*NorESM model description*

The NorESM simulations are carried out with the NorESM1-F version of the model, which is described in detail in Guo

et al. (2019). The NorESM version that contributes to CMIP5 is NorESM1-M. It has a ∼2° resolution atmosphere and land configuration, and a nominal 1° ocean and sea ice configuration. In NorESM1-F, the same atmosphere–land grid is used as NorESM1-M (CMIP5 version), whereas a tripolar grid is used for the ocean–sea ice components in NorESM1-F, instead of the bipolar grid in NorESM1-M. The tripolar grid is also used in the CMIP6 version of NorESM (NorESM2). NorESM1-F runs about 2.5 times faster than NorESM1-M. For the preindustrial, NorESM-F has a more realistic Atlantic meridional overturning

circulation than NorESM1-M.

*NorESM model simulations*

The NorESM simulations are carried out at ×2, and ×4 $CO_2$ concentrations. The ocean temperatures were initialized from the ×2 $CO_2$ Eocene simulations with the lower resolution NorESM-L model (Zhang et al., 2012). The ocean salinity was initialized with constant values of 25.5 psu in the Arctic and 34.5 psu elsewhere. From the initial conditions, the ×2 $CO_2$

experiment was in total run for 2100 years. The ×4 $CO_2$ was branched from the end of the 100th year of ×2 $CO_2$ experiment, and run for 2000 years. The results from the last 100 years were used in the study. Note that the NorESM simulations were carried out with the Baatsen et al. (2016) paleogeography (based on a paleomagnetic reference frame), not the Herold et al. (2014) paleogeography (based on a mantle reference frame), in contrast to the other simulations described in this paper.

## 3  Results

We discuss the results from the model simulations, focusing on the model spinup and equilibrium (Section 3.1) followed by three aspects which align with the research questions outlined at the end of Section 1: the large-scale features of the modelled temperature response compared with preindustrial (Section 3.2), the reasons for the different model responses (Section 3.3), and a comparison with paleo proxy data (Section 3.4).

### 3.1  Model spinup and equilibrium

It is important to assess to what extent the Eocene simulations represent an equilibrated state. This is because for many models the initial condition may be far from the ultimate equilibrium, and as such very long simulations are required to reach this equilibrium, which may be prohibitive in terms of computation and time resource. For all the DeepMIP simulations, the length, and top of atmosphere (TOA) inbalance and near-surface global mean air temperature trend at the end of the simulation, are summarised in Table S1 in the Supplement. The TOA inbalance and temperature trends are also given for the associated

preindustrial simulations. As part of the DeepMIP experimental design (Lunt et al., 2017), formulated prior to any simulations had started running, it was suggested that appropriate criteria for sufficient model equilibration would be that simulations should ideally be "(a) at least 1000 years in length, and (b) have an imbalance in the top-of-atmosphere net radiation of less than $0.3Wm^{-2}$ (or have a similar imbalance to that of the preindustrial control), and (c) have sea-surface temperatures that are not strongly trending (less than 0.1 °C per century in the global mean).". All the simulations satisfy criterion (a). All simulations

except for CESM (×3, ×6 and ×9) and IPSL (×1.5 and ×3) satisfy criterion (b). Note that for some models, the preindustrial TOA inbalance is relatively large; this may be due to non-conservation (e.g. COSMOS; Stevens et al., 2013) or due to some

energy fluxes being calculated at the top-of-the-model rather than top-of-the-atmosphere (e.g. INMCM); in these cases the TOA inbalance is not a good diagnostic for equilibration because there is some atmosphere above the top of the model which can interact with incoming or outgoing radiation (i.e. the model top is not at 0 mbar). All the models except for CESM ($\times 3$),

COSMOS ($\times 4$), and HadCM3 ($\times 2$ and $\times 3$) satisfy criterion (c). Overall, all models satisfy at least two of the three criteria, except for CESM at $\times 3$ which is nonetheless close to both missed criteria (0.32 versus 0.30 Wm$^{-2}$ and 1.1 versus 1.0$^{\circ}$C). As such, we make a decision to accept all simulations as being sufficiently equilibrated to be included in the ensemble, but note that further spinup would be required to confirm the results of those simulations with relatively large residual trends or anomalous TOA inbalances.

It is also worth noting that some models crashed when run under $CO_2$ concentrations higher than in the simulations described here. In particular, CESM crashed at $\times 12$, COSMOS crashed at $\times 6$, HadCM3 crashed at $\times 4$, IPSL crashed at $\times 6$, and MIROC crashed at $\times 4$. These crashes have not been explored in detail, but could be due to feedbacks becoming more positive as temperature increases (for example associated with an increase in height of the tropopause; Meraner et al., 2013), to such an extent that positive feedbacks overcome the negative Planck feedback (Bloch-Johnson et al., 2015), at which point a "runaway"

phase is entered and the temperature begins to increase rapidly. This can then cause violation of the CFL criterion due to high wind speeds associated with the generation of large pressure and/or temperature gradients, causing the model to crash.

## 3.2    Documentation of large-scale features

Here we present the large-scale features of the DeepMIP simulations, with a focus on annual mean temperature. We start with global mean quantities, move on to latitudinal gradients, and finish by describing the spatial patterns.

Figure 1a shows the global mean near-surface air temperature as a function of model $CO_2$ for each DeepMIP simulation and associated preindustrial control, plus some previous Eocene simulations carried out with other boundary conditions (Lunt et al., 2012; Kiehl and Shields, 2013; Sagoo et al., 2013). The DeepMIP simulations are fairly consistent in terms of global mean temperature for a given $CO_2$ concentration across the ensemble. The exception to this is INMCM, which at $\times 6$ $CO_2$ has a lower global mean temperature than any of the $\times 3$ simulations. This is consistent with the fact that of all the models

in the CMIP6 ensemble INMCM has the lowest climate sensitivity (Zelinka et al., 2020). Excepting INMCM, the spread in the DeepMIP simulations is substantially less than in the previous Eocene simulations. In particular, at $\times 3$ $CO_2$, the CESM, COSMOS, GFDL, HadCM3, IPSL, and MIROC simulations are within 1.9$^{\circ}$C, compared with 5.0$^{\circ}$C at $\times 4$ for the previous simulations. Part of the reason for the reduced spread of many of the DeepMIP simulations compared with previous simulations may be related to the fact that all the DeepMIP model simulations have the same prescribed paleogeography, land-sea mask,

and vegetation, whereas previous simulations used a variety of these boundary conditions.

The DeepMIP models have a range of Eocene climate sensitivities to $CO_2$ doubling; from a minimum of 2.9$^{\circ}$C (for NorESM) to a maximum of 5.6$^{\circ}$C (for IPSL, excluding the anomalously warm $\times 9$ CESM simulation). The average of the DeepMIP climate sensitivities (again excluding the $\times 9$ CESM simulation) is 4.5$^{\circ}$C, which is greater than the average of the previous simulations (3.3 $^{\circ}$C). There is a non-linearity (i.e. a global mean temperature that increases with $CO_2$ differently than would be

expected from a purely logarithmic relationship) in the CESM model simulations (as previously noted by Zhu et al., 2019), and

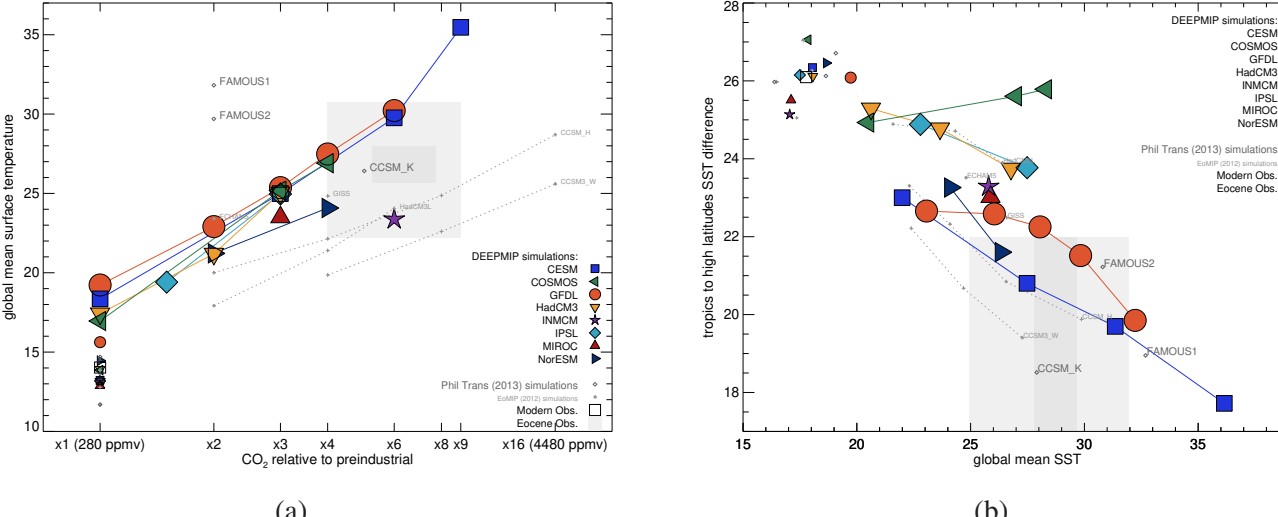

**Figure 1.** (a) Global annual mean near-surface (2m) air temperature in the DeepMIP simulations, as a function of atmospheric $CO_2$. Large coloured symbols show the Eocene simulations, and smaller coloured symbols show the associated preindustrial controls. Also shown are results from some previous Eocene simulations (Lunt et al., 2012; Kiehl and Shields, 2013; Sagoo et al., 2013) and associated preindustrial control simulations (small grey symbols). The models that have carried out Eocene simulations at more than one $CO_2$ concentration are joined by a straight line. Open square shows modern observations. The grey filled boxes show estimates of the global mean temperature (from Inglis et al., 2020) and $CO_2$ (from Anagnostou et al., 2020) derived from proxies. For temperature, the light grey box shows the 10 to 90% confidence interval and the dark grey box shows the 33 to 66% confidence interval; for $CO_2$, the light grey box shows $\pm 1$ s.d and the dark grey box shows $\pm 2$ s.d.; see Section 3.4 for more details. (b) As (a), but for meridional SST gradient as a function of global mean SST. Meridional SST gradient is defined here as the average SST equatorwards of $\pm 30°$ minus the average SST polewards of $\pm 60°$. The grey filled boxes show estimates of the global mean SST (from Inglis et al., 2020) and SST gradient (from Cramwinckel et al., 2018; Evans et al., 2018; Zhu et al., 2019) derived from proxies. For SST, the light grey box shows the 10 to 90% confidence interval and the dark grey box shows the 33 to 66% confidence interval; for meridional temperature gradient, the light grey box shows the range (which extends below the y-axis limit, down to $14°$C); see Section 3.4 for more details.

also in HadCM3 and (to a lesser extent) GFDL and COSMOS. In CESM the climate sensitivity, normalised to a $CO_2$ doubling, increases from $4.2°$C at $\times 1$ to $4.8°$C and $9.7°$C at $\times 3$ and $\times 6$, respectively. In GFDL the climate sensitivity increases from $3.7°$C at $\times 1$ to $5.1°$C at $\times 3$, but then decreases to $4.7°$C at $\times 4$. In HadCM3 the climate sensitivity increases from $3.8°$C at $\times 1$ to $6.6°$C at $\times 2$. In COSMOS the climate sensitivity decreases from $5.2°$C at $\times 1$ to $4.2°$C at $\times 3$. In CESM, the non-linearity has been shown to arise from an increase in strength of the positive shortwave cloud feedback as a function of temperature (Zhu et al., 2019), and is most apparent in the transition from $\times 6$ to $\times 9$.

CESM, COSMOS, GFDL, and HadCM3 all carried out simulations at $\times 1$ $CO_2$; comparison with the associated preindustrial controls indicates that the non-$CO_2$ component of global warmth (i.e. due to changes paleogeography, vegetation, and aerosols, and removal of continental ice sheets) is $5.1°$C, $3.6°$C, $3.5°$C, and $3.1°$C for CESM, GFDL, HadCM3, and COSMOS

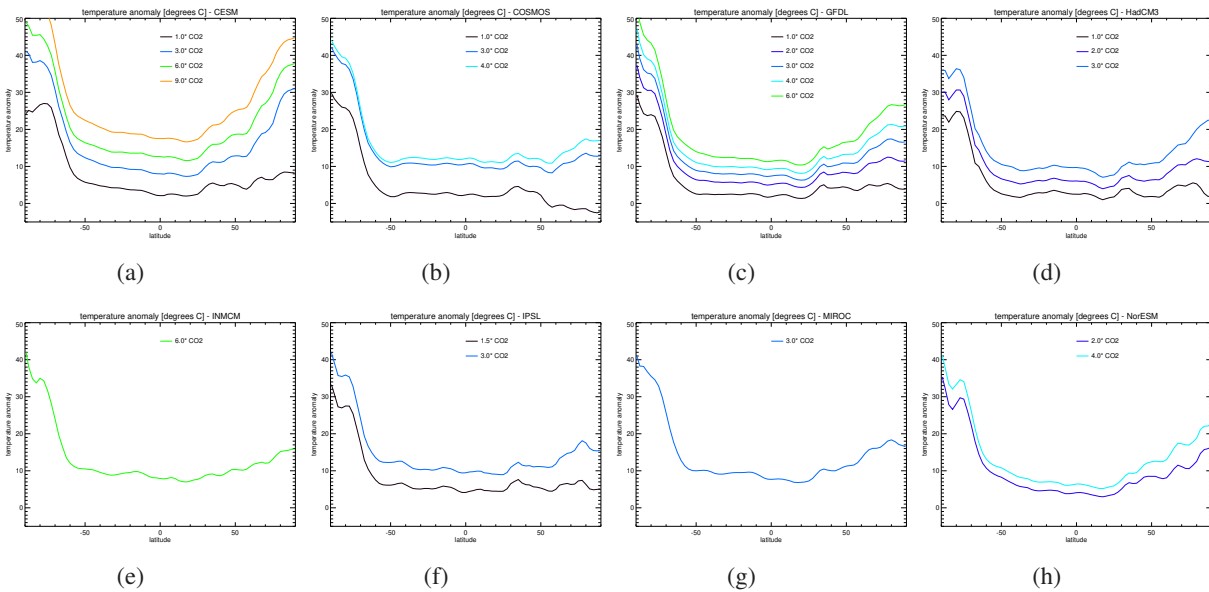

**Figure 2.** Zonal mean near-surface air temperatures in the DeepMIP simulations, as a function of latitude and prescribed atmospheric $CO_2$ concentration, expressed as anomalies relative to the equivalent preindustrial control. (a) CESM, (b) COSMOS, (c) GFDL, (d) HadCM3, (e) INMCM, (f) IPSL, (g) MIROC, and (h) NorESM.

respectively. This is for comparison with previous simulations using CCSM3 (Caballero and Huber, 2013) which indicated a non-$CO_2$ warming of $\sim 5°C$.

The latitudinal gradient of SST, defined here as the average SST equatorwards of $\pm 30°$ minus the average temperature polewards of $\pm 60°$, is shown in Figure 1b. All DeepMIP models that have carried out simulations at more than one $CO_2$ concentration show a decrease in meridional SST gradient as temperature increases, apart from COSMOS. COSMOS also has the

strongest preindustrial meridional temperature gradient. The $\times 1$ $CO_2$ Eocene simulations indicate that the non-$CO_2$ DeepMIP boundary conditions decrease the latitudinal gradient by $3.4°C$ for GFDL, $3.3°C$ for CESM, $2.1°$ for COSMOS, and $0.8°C$ for HadCM3. The GFDL model displays a markedly non-linear response, with a more rapidly decreasing temperature gradient as a function of temperature at higher temperatures than at lower temperatures. In contrast to the global mean temperature, the DeepMIP models show substantial spread in meridional temperature gradient across the ensemble; COSMOS has a particu-

larly strong gradient in the Eocene at $\times 3$ and $\times 4$ $CO_2$, and HadCM3 and IPSL also have relatively strong gradients, similar to previous Eocene simulations with HadCM3L (Lunt et al., 2010b).

The zonal mean near-surface air temperature anomaly, relative to preindustrial, as a function of latitude is shown in Figure 2. Polar amplification is clear in both hemispheres for all models at $CO_2 > \times 1$. There is greater amplification in the Southern Hemisphere than in the Northern, due to the replacement of the Antarctic ice sheets with vegetated land surface, with associated

local warming due to the altitude and albedo change. There is a similar pattern of response across the models for a given $CO_2$ concentration. However, although the models have a similar response in the Southern Hemisphere, the CESM model has greater

polar amplification than other models in the Northern Hemisphere for a given $CO_2$ concentration (in particular at $\times 3$ $CO_2$). The pattern of warming in the $\times 1$ simulations is similar between the CESM, GFDL, and HadCM3 models. In particular, they all exhibit warming around 30-40° North, which coincides with lower topography in the Tibetan plateau region in the Eocene relative to preindustrial. There is also consistent warming in the Northern Hemisphere Arctic (excepting COSMOS) which coincides with the absence of the Greenland ice sheet and boreal forest in place of tundra and bare soil in the preindustrial. The same underlying structure is seen in the higher $CO_2$ simulations (see for example in GFDL, Figure 2b).

The spatial pattern of surface air temperature response is shown in Figure 3. Because of the difference in continental positions between the preindustrial and Eocene, we show the difference between the Eocene and the zonal mean of the preindustrial, i.e. $GAT_e^m - \overline{GAT_p^m}$ in the notation of Lunt et al. (2012). This shows some consistent responses across the ensemble. In particular, in addition to the polar amplification, the response is characterised by greater warming over land than over ocean. Many of those continental regions where the warming is more muted (such as the Rockies, tropical east Africa, India, and the mid-latitudes of East Asia) are associated with regions of high topography in the Eocene. There is also substantial warming in the North Pacific in all simulations. This may be associated with deep water formation in this region driving poleward heat transport in the Pacific, but the ocean circulation in these simulations will be explored in a subsequent study.

A similar plot, but without the zonal mean of the preindustrial (i.e. $GAT_e^m - GAT_p^m$), is shown in the Supplement, Figure S1. Figure S1 also includes the Eocene simulations at $\times 1$ and $\times 1.5$. The Eocene $\times 1$ simulations minus preindustrial show the spatial impact of the changes to the non-$CO_2$ boundary conditions. Consistent across the ensemble is the clear warming in Antarctica associated with the altitude and albedo change, warming in the Tibetan plateau associated with altitude change, and cooling in Europe.

## 3.3  Reasons for model spread

Here we first qualitatively explore the different model results, by presenting the changes in albedo and emissivity across the ensemble. We then quantitatively relate these to the zonal-mean temperature change, and global metrics, by making use of a 1-dimensional energy-balance framework. Future work in the framework of DeepMIP will explore the model simulations in more detail, in particular the response of clouds, the hydrological cycle, and ocean circulation.

The patterns of surface albedo in the preindustrial and Eocene simulations are shown in the Supplement, Figure S2. The lower albedo associated with the lack of Antarctic ice sheet in the Eocene is clear for all the models. In addition, the Eocene models do not have the high albedo associated with modern subtropical deserts (the Eocene experimental design specified average soil properties to be prescribed for all non-vegetated surfaces). The gradual decrease in high latitude albedo with increasing surface temperature is apparent in all models, over both land and ocean, due to decreasing snow and sea-ice cover. GFDL has a relatively low albedo prescribed over land in the preindustrial, which is consistent with its relatively warm global mean (Figure 1(a); small red circle). CESM in general retains more snow cover than other models over Antarctica for a given $CO_2$ concentration. NorESM has a relatively low prescribed albedo over land in the Eocene. The patterns of planetary albedo in the preindustrial and Eocene simulations are shown in the Supplement, Figure S3. Again, the high albedo over high latitude regions is clear, although the planetary albedo over Antarctica in the preindustrial is lower then the surface albedo, indicating

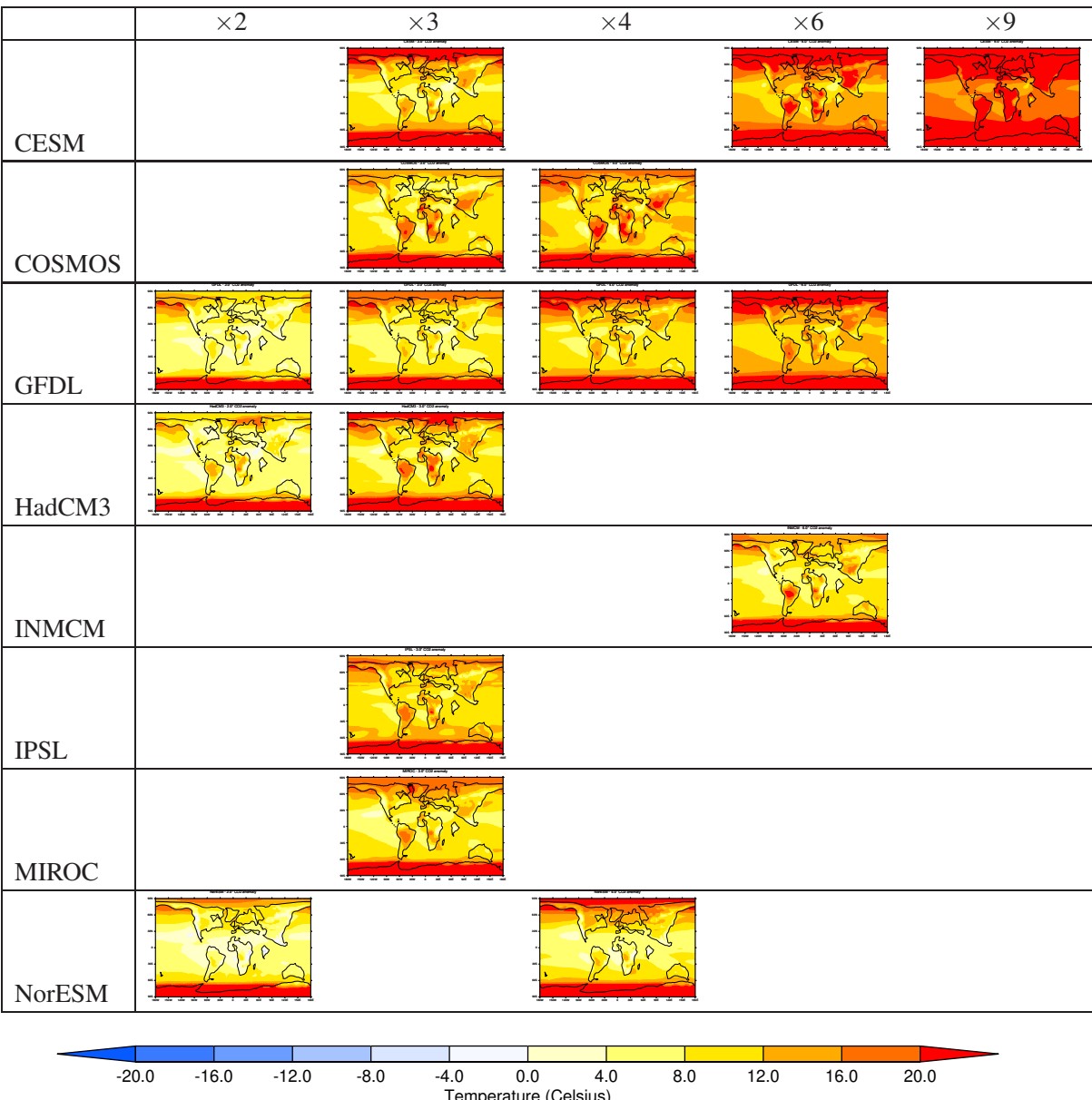

**Figure 3.** DeepMIP near-surface air temperature anomalies, relative to the zonal mean of the associated preindustrial simulation, ordered by $CO_2$ concentration and by model. Simulations with $CO_2$ equal or greater than $\times 2$ are shown. The variable plotted is $\mathrm{GAT}_e^m - \overline{\mathrm{GAT}_p^m}$ in the notation of Lunt et al. (2012).

that the presence of clouds lowers the albedo in this region. Globally there is a transition to lower values as temperature increases, and the regions associated with the lowest values (e.g. the subtropics in CESM) tend to expand in area, associated with decreases of cloud cover and opacity (Zhu et al., 2019). However, GFDL retains a high planetary albedo in the Arctic even

at $\times 6$ $CO_2$, despite a low surface albedo, indicating persistent cloud cover in this region. MIROC appears to have less spatial
structure in planetary albedo than the other models. The patterns of emissivity in the preindustrial and Eocene simulations
are shown in the Supplement, Figure S4. The relatively low emissivity associated with the high altitude Antarctic ice sheet in
the modern is apparent. The emissivity decreases in general as temperature increases, likely associated with increasing water
vapour and changes in clouds, with the patterns remaining fairly consistent as temperature increases, with the lowest values
over the warm pool in the western tropical Pacific.

In order to quantitatively relate these differences in radiative fluxes to the differences in temperature presented in Section 3.2,
we make use of the energy-balance framework described in Heinemann et al. (2009), and used previously to explore Eocene
simulations by Lunt et al. (2012). In this framework, the zonal mean surface temperature ($\tau$), planetary albedo ($\alpha_p$), emissivity
($\epsilon$), incoming TOA solar radiation ($S$), and meridional heat flux ($H$), are related by

$$S(1 - \alpha_p) + H = \epsilon \sigma \tau^4, \tag{1}$$

where $\sigma$ is the Stefan-Bolzmann constant, and where $\alpha_p$, $\epsilon$, $H$, and $S$ are functions of latitude that can be derived from the
modelled energy fluxes, from either the preindustrial ($x^{P1}$) or $\times$N $CO_2$ Eocene ($x^{EN}$) simulations. In our case, the solar
constant is the same in the preindustrial and Eocene simulations, and so by rearranging Equation 1 we can write $\tau$ as a function
of $\alpha_p$, $\epsilon$ and $H$. For example, the surface temperature of the standard Eocene $\times 3$ simulation is $\tau(\alpha_p^{E3}, \epsilon^{E3}, H^{E3})$, and that of a
preindustrial simulation is $\tau(\alpha_p^{P1}, \epsilon^{P1}, H^{P1})$. The contribution of emissivity changes to the Eocene warming at $\times 3$ relative to
preindustrial, $\Delta\tau_\epsilon$ is then given by $\tau(\alpha_p^{P1}, \epsilon^{E3}, H^{P1}) - \tau(\alpha_p^{P1}, \epsilon^{P1}, H^{P1})$, and similarly for meridional heat flux and planetary
albedo:

$$
\begin{aligned}
\Delta\tau_\epsilon &= \tau(\alpha_p^{P1}, \epsilon^{E3}, H^{P1}) - \tau(\alpha_p^{P1}, \epsilon^{P1}, H^{P1}) \\
\Delta\tau_H &= \tau(\alpha_p^{P1}, \epsilon^{P1}, H^{E3}) - \tau(\alpha_p^{P1}, \epsilon^{P1}, H^{P1}) \\
\Delta\tau_{\alpha_p} &= \tau(\alpha_p^{E3}, \epsilon^{P1}, H^{P1}) - \tau(\alpha_p^{P1}, \epsilon^{P1}, H^{P1})
\end{aligned} \tag{2}
$$

Heinemann et al. (2009) and Lunt et al. (2012) showed how this framework could be expanded to also include terms related
to longwave and shortwave cloud changes, by including terms derived from the clear-sky fluxes from the model radiation
scheme. Here we choose instead to partition the planetary albedo term ($\Delta\tau_{\alpha_p}$) into a surface albedo term ($\Delta\tau_{\alpha_s}$) and a non-
surface albedo term ($\Delta\tau_{\alpha_{ns}}$), as such:

$$
\begin{aligned}
\Delta\tau_{\alpha_s} &= \tau(\alpha_p^{P1} + (\alpha_s^{E3} - \alpha_s^{P1}), \epsilon^{P1}, H^{P1}) - \tau(\alpha_p^{P1}, \epsilon^{P1}, H^{P1}) \\
\Delta\tau_{\alpha_{ns}} &= \Delta\tau_{\alpha_p} - \Delta\tau_{\alpha_s}
\end{aligned} \tag{3}
$$

where $\alpha_s$ is the surface albedo. The surface albedo changes are a result of prescribed vegetation and ice sheet albedo changes,
and snow and sea-ice feedbacks. The non-surface albedo changes are a result of cloud and aerosol changes, or cloud masking
effects (see below). Note that due to the non-linear dependence of albedo and emissivity on the radiative fluxes, the results are
sensitive to the order of zonal mean, annual mean, and albedo/emissivity operators, but this has a generally small effect, except
in the partitioning of surface and non-surface albedo in the high latitudes where it can have an effect of $\pm 3°C$ (not shown).

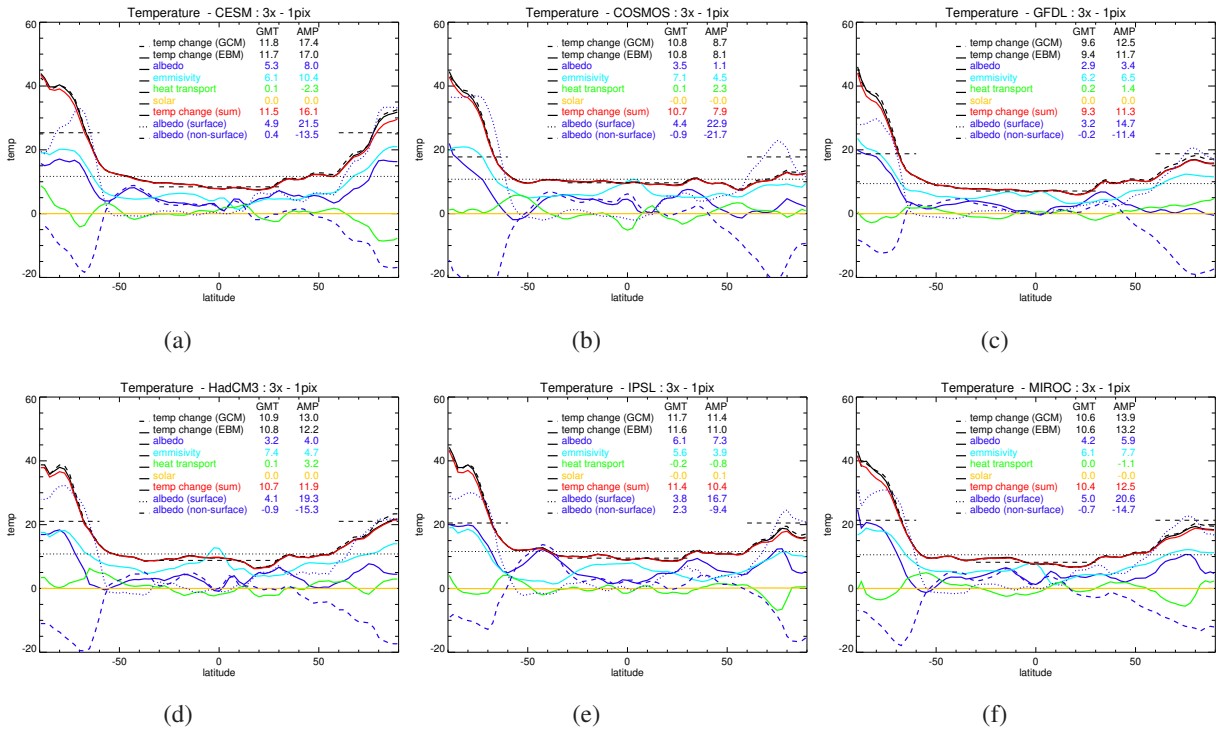

**Figure 4.** The results of the energy balance analysis as described in Equations 2 and 3, applied to the differences between the DeepMIP ×3 simulations and their associated preindustrial controls. Black dashed line shows the zonal mean surface temperature changes directly from the GCMs. Black solid line shows the temperature change derived from the radiative fluxes, $\Delta\tau$. Solid blue, cyan and green lines show the contributions from planetary albedo ($\Delta\tau_\epsilon$), emissivity ($\Delta\tau_\epsilon$), and meridional heat flux ($\Delta\tau_\epsilon$) respectively (Equation 2). Blue dotted and dashed lines show the contribution from surface albedo ($\Delta\tau_{\alpha_s}$) and non-surface albedo ($\Delta\tau_{\alpha_{ns}}$) respectively (Equation 3). The red line shows the sum of the individual terms. For each model, the contribution of each term to the changes in global mean temperature (GMT), and polar amplification (AMP; expressed as the the difference in warming between the high latitudes (polewards of $\pm60°$) and the tropics ($\pm30°$)), is quantified in the legend.

The results of this analysis are shown in Figure 4, for those models that carried out ×3 simulations (all models except for INMCM and NorESM). This shows that in general all models have similar reasons for their response to the DeepMIP boundary conditions. In particular, in the equatorial region (latitudes $\pm10°$), the temperature response is in general dominated by emissivity changes, and in the subtropics is dominated by emissivity and albedo (specifically, non-surface albedo) changes. In southern hemisphere high latitudes, both emissivity and albedo changes contribute to warming. The change in altitude over Antarctica is likely a large part of this emissivity contribution. The albedo-induced change is made up of a large positive surface albedo contribution which is partially cancelled by a negative non-surface albedo contribution. This partial cancellation is a result of the very strong surface albedo change over Antarctica. In the absence of clouds, this surface albedo change on its own would cause large changes in temperature. However, in reality, some of these changes are masked by clouds and as such do not

have as big an effect as would be the case in a cloud-free state. In the Northern Hemisphere, the signals are more variable across the ensemble. Most models show similar behaviour to the Southern Hemisphere, with positive contributions from emissivity and surface albedo, and a negative contribution from non-surface albedo (again resulting from the cloud masking effect, over the Arctic sea-ice). However, in COSMOS and GFDL the Arctic response is dominated by emissivity changes, with relatively little contribution from albedo.

The global mean warming, $\times 3$ minus preindustrial, is fairly constant across the ensemble. The greatest warming is in CESM (11.8°C), for which 6.1°C comes from emissivity and 5.3°C comes from albedo (4.9°C from surface albedo and 0.4°C from non-surface albedo). The least warming is in GFDL (9.6°C) for which 6.2°C comes from emissivity and 2.9°C comes from albedo (3.2°C from surface albedo and -0.2°C from non-surface albedo). Therefore, the difference in sensitivity between these two end-members of the ensemble primarily results from reduced surface albedo change in GFDL compared with CESM, and secondarily from negative non-surface albedo changes in GFDL compared with positive in CESM.

The reasons for the polar amplification are more variable between the models. For the model with the greatest polar amplification, CESM (17.4°C), this is made up of 8.0°C from albedo, 10.4°C from emissivity, and -2.3°C from meridional heat flux. For the model with the least polar amplification, COSMOS (8.7 °C), this is made up of 1.1°C from albedo, 4.5°C from emissivity, and 2.3°C from meridional heat flux. Other models share relatively similar polar amplification (ranging from 11.4°C in IPSL to 13.9°C in MIROC), but the reasons for this vary between the models; in IPSL the dominant contribution is from albedo, in GFDL it is from emissivity with a positive contribution from meridional heat flux, in MIROC it is also from emissivity but with a negative contribution from meridional heat flux, and in HadCM3 it is roughly equal between albedo and emissivity, with a strong contribution also from meridional heat flux.

The differences above, $\times 3$ minus preindustrial, can be considered as consisting of a component due to non-$CO_2$ boundary condition changes ($\times 1$ minus preindustrial) and a component due to $CO_2$ change ($\times 3$ minus $\times 1$). Four of the models (CESM, COSMOS, GFDL, and HadCM3) also carried out simulations at $\times 1$ which allow us to diagnose this partitioning. The energy-balance analysis for $\times 1$ minus preindustrial and $\times 3$ minus $\times 1$ is shown in Figures S5 and S6 of the Supplement (note that due to non-linearities the sum of these two partitions does not exactly equal the $\times 3$ minus preindustrial values shown in Figure 4). This shows that the non-$CO_2$ response (Supplement, Figure S5) is greatest in the polar regions of the Southern Hemisphere, where albedo and emissivity contribute approximately equally in all models. Elsewhere, the signal is small; for the global mean, albedo and emissivity contribute roughly equally, although in CESM albedo dominates and in GFDL emissivity dominates. For the $CO_2$-only response (Supplement, Figure S6), on a global scale emissivity changes dominate in all models. As expected, the contribution due to surface albedo changes is close to zero in all regions except the high latitudes. All models show polar amplification in both hemispheres, but the reasons for this vary. CESM polar amplification is due to both emissivity and albedo changes, and is offset by changes in meridional heat flux whereas the other models are dominated by emissivity and meridional heat flux changes, and offset by albedo changes (due to strong offsetting by non-surface albedo). The importance of changes in outgoing long-wave radiation for polar amplification in response to $CO_2$ forcing is also seen in model simulations of the modern climate (Pithan and Mauritsen, 2014).

**Table 2.** Summary of the contributions to global mean surface warming and polar amplification from preindustrial to ×3. Values are shown for the 4 DeepMIP models that carried out simulations of the preindustrial, ×1, and ×3 (CESM, COSMOS, GFDL, and HadCM3). The values correspond to those shown in Figures S5 and S6 in the Supplement. Note that due to non-linearities the sum of these is slightly different from the values in Figure 4. Polar amplification is defined as the difference in warming between the high latitudes (polewards of $\pm 60°$) and the tropics ($\pm 30°$).

| Variable [°C] | CESM | COSMOS | GFDL | HadCM3 | 4-model mean |
|---|---|---|---|---|---|
| Global mean surface warming | 11.5 | 10.6 | 9.2 | 10.7 | 10.5 |
| Emissivity (×1-PI) | 1.7 | 1.5 | 1.9 | 1.7 | 1.7 |
| Surface albedo (×1-PI) | 3.8 | 3.9 | 2.0 | 3.3 | 3.3 |
| Non-surface albedo (×1-PI) | -0.4 | -2.4 | -0.5 | -1.7 | -1.3 |
| Emissivity (×3-×1) | 4.5 | 5.6 | 4.3 | 5.8 | 5.1 |
| Surface albedo (×3-×1) | 1.1 | 0.6 | 1.2 | 0.8 | 0.9 |
| Non-surface albedo (×3-×1) | 0.8 | 1.5 | 0.2 | 0.8 | 0.8 |
| Polar amplification | 16.3 | 7.9 | 11.5 | 11.9 | 11.9 |
| Emissivity (×1-PI) | 7.4 | 3.7 | 3.5 | 4.1 | 4.7 |
| Surface albedo (×1-PI) | 14.6 | 20.1 | 8.0 | 15.8 | 14.6 |
| Non-surface albedo (×1-PI) | -9.2 | -17.0 | -3.8 | -11.4 | -10.3 |
| Meridional heat flux (×1-PI) | -1.9 | -1.9 | 0.1 | -0.6 | -1.1 |
| Emissivity (×3-×1) | 3.1 | 0.8 | 3.2 | 0.7 | 1.9 |
| Surface albedo (×3-×1) | 7.6 | 3.2 | 7.2 | 4.0 | 5.5 |
| Non-surface albedo (×3-×1) | -4.8 | -5.2 | -8.0 | -4.4 | -5.6 |
| Meridional heat flux (×3-×1) | -0.4 | 4.2 | 1.3 | 3.6 | 2.2 |

By way of summary, the reasons for the difference ×3 minus preindustrial, for the 4 models for which we can carry out a full partitioning, are given in Table 2. This shows that averaged across these 4 models, of the $\sim 10°C$ warming, about 5°C arises from emissivity changes from the $CO_2$ increase (and associated water vapour and longwave cloud feedbacks), about 2°C arises from albedo changes from the non-$CO_2$ boundary conditions, primarily removal of ice and changes in vegetation and aerosols (and associated cloud, snow, and sea-ice feedbacks), about 1.5°C arises from emissivity changes from the non-$CO_2$ boundary conditions, primarily lower Antarctic altitude (and associated water vapour changes), and about 1.5°C arises from albedo changes from the $CO_2$ increase, i.e. cloud, snow, and sea-ice feedbacks. For polar amplification, of the $\sim 12°C$, about 5°C arises from the emissivity changes from the non-$CO_2$ boundary conditions, about 4°C arises from the albedo changes from the non-$CO_2$ boundary conditions, about 2°C arises from emissivity changes from the $CO_2$ increase, and about 1°C arises from heat flux changes (made up of a contribution of +2°C from the $CO_2$ increase and -1°C from the non-$CO_2$ changes).

### 3.4 Model-data comparison

Here we present a comparison of the models with proxy data of Eocene temperature and $CO_2$. After introducing the proxy datasets we compare the models to proxy-based global metrics, and then to specific locations on a point-point basis.

#### 3.4.1 Proxy datasets

For point-to-point model-data comparisons, we use the SST and surface air temperature (SAT) datasets for the EECO compiled by Hollis et al. (2019). Following their recommendation, we exclude $\delta^{18}$O-derived SST estimates from recrystallized planktonic foraminifera because these estimates are in general significantly cooler than estimates derived from the $\delta^{18}$O value of well-preserved foraminifera, foraminiferal Mg/Ca ratios, and clumped isotope values from larger benthic foraminifera, due to diagenetic effects.

In terms of global metrics, we make use of the global mean near-surface air temperature (GSAT) estimate for the EECO from Inglis et al. (2020), which is based on the Hollis et al. (2019) temperature dataset, and also excludes SST estimates from recrystallised foraminifera. The vertical dimensions of the grey filled boxes in Figure 1(a) show the 33 to 66% and 10 to 90% confidence intervals of this GSAT estimate. For global mean SST, we again use the GSAT estimate of Inglis et al. (2020) but convert this to global mean SST using a linear function derived from the mean land-sea temperature contrast in all the model simulations shown in Figure 1:

$$SST = 0.82 \times GSAT + 6.6^{\circ}C. \tag{4}$$

This SST estimate forms the horizontal dimensions of the grey filled boxes in Figure 1(b).

We make use of SST gradient estimates from Cramwinckel et al. (2018), Evans et al. (2018), and Zhu et al. (2019). Cramwinckel et al. (2018) define a meridional temperature gradient metric as the difference between tropical mean SST (derived from $TEX_{86}$) and deep-ocean temperatures (derived from $\delta^{18}$O), assuming that deep ocean temperatures are an approximation to high-latitude SSTs. In this way they reconstruct a metric at 50 Ma of about 20-22$^{\circ}$C (their Figure 3b). Evans et al. (2018) use a similar approach but using tropical SSTs and deep-ocean temperatures from Mg/Ca between 48 and 56 Ma, and find a reduction in their metric from the modern to early Eocene of 22% to 42%. Given a modern gradient from HadISST (our Figure 1b) of about 26$^{\circ}$C, this gives an early Eocene estimate of 17.7$^{\circ}$C (20.3 to 15.1$^{\circ}$C). Using a similar approach to Evans et al. (2018), Zhu et al. (2019) find a reduction in their metric of about 20% to 45% (their Figure 1b), giving an estimate of 21 to 14$^{\circ}$C for the early Eocene. Given the differences in methodologies for deriving these estimates, the relatively wide time window of "early Eocene" for two of the studies, and the differences between the proxy metrics and our modelled metric (defined here as the average SST equatorwards of $\pm30^{\circ}$ minus the average SST polewards of $\pm60^{\circ}$), we take as our overall proxy estimate the outer ranges of the three studies, giving a range of 14 to 22$^{\circ}$C. This range in meridional temperature gradient estimate forms the vertical edge of the grey filled boxes in Figure 1(b). However, the use of benthic temperatures to approximate high latitude annual mean surface temperature may result in biases due to the seasonality of deep water formation (Evans et al., 2018), and we note that a detailed assessment of the meridional temperature gradient implied by proxies, similar to that carried out for global mean temperature by Inglis et al. (2020), would be beneficial for future model-data comparisons.

For $CO_2$, Anagnostou et al. (2020) give two estimates of EECO $CO_2$ based on two different calibrations, resulting in 95% confidence intervals of 1170 to 1830 ppmv and 1540 to 2490 ppmv. The uppermost and lowermost bounds of these two estimates are close to the ×4 (1120 ppmv) and ×9 (2520 ppmv) simulations; as such, these form the horizontal edges of the light grey box in Figure 1. A normal distribution in absolute $CO_2$ would give a corresponding 68% confidence interval of 1470 to 2170 ppmv, and this forms the horizontal edges of the light grey box in Figure 1.

Overall, for the purposes of describing the model-data consistency, we use "*consistent*" to describe a model that sits within the light grey boxes of Figure 1(a,b), and "*very consistent*" to describe a model that sits within the dark grey boxes of Figure 1(a,b).

### 3.4.2 Comparison with global metrics

For the DeepMIP models, only those that carried out simulations at ×4, ×6 $CO_2$, or ×9 $CO_2$ are consistent with the $CO_2$ proxies (i.e. CESM, COSMOS, GFDL, INMCM, and NorESM), and only those that carried out simulations at ×6 $CO_2$ are very consistent with the $CO_2$ proxies (i.e. CESM, GFDL, and INMCM). All these simulations are also consistent with the GSAT proxies, but only COSMOS and GFDL at ×4 are very consistent with the GSAT proxies (inspection of Figure 1 indicates that CESM would also be very consistent with the GSAT proxies if there was a simulation at ×4). No simulations are very consistent with both the $CO_2$ and GSAT proxies. Only CESM at ×6, GFDL at ×4 and ×6, and NorESM at ×4 are consistent with the $CO_2$, GSAT, and meridional temperature gradient proxies.

Of the pre-DeepMIP simulations in Figure 1(a,b), CCSM_H and CCSM_W at ×8 are also consistent with all the proxy constraints, and CCSM_K is additionally very consistent with GSAT. However, as discussed in Section 1, CCSM_K includes somewhat arbitrary modifications to cloud parameters that are designed to enable the model to better fit the Eocene observations, and as such, in contrast to the DeepMIP simulations, cannot be considered entirely independent from the temperature data.

Some quantitative metrics for the simulations are presented in the Supplement, Table S2. In this case, the metrics are given for the set of simulations that were carried out at $CO_2$ concentrations consistent with the proxies.

### 3.4.3 Comparison with specific locations

The limited range of $CO_2$ concentrations explored by some models, coupled with the relatively large uncertainties in EECO $CO_2$ from proxies, means that a model-data comparison of individual model simulations with the site-by-site proxy data can be misleading. As such, here we only carry out a detailed model-data comparison for those models that have carried out simulations under more than one $CO_2$ concentration. For those models (CESM, COSMOS, GFDL, HadCM3, IPSL, and NorESM), we apply a global-mean scaling factor to the simulated SST and GSAT such that the modelled global means best fit the global mean proxy data. We then compare the spatial patterns in the scaled model outputs with the spatial patterns in the site-by-site proxies. We provide a quantitative metric for the model-data fit, and compare this with some idealised temperature distributions to put these metrics in context.

We scale the models by assuming that the spatial pattern of temperature change scales linearly with global mean temperature change, and by interpolating or extrapolating to a global mean equal to the estimate from Inglis et al. (2020), i.e. 27°C for near-surface air temperature, or an equivalent global mean SST given by Equation 4. This process gives a scaling factor, $s$ that can be used to create a spatial field of implied temperature, $T^i$ that is consistent with this proxy-based global mean temperature, $< T^p >$:

$$s = \frac{< T^p > - <^L T >}{<^H T > - <^L T >}$$

$$T^i = s(^H T - ^L T), \tag{5}$$

where $^L T$ and $^H T$ are the spatial fields of the two model simulations that have global means closest to $< T^p >$, and where $<>$ denotes a global mean quantity. This also allows us to calculate an inferred $CO_2$ concentration, $CO_2^i$:

$$CO_2^i = ^L CO_2 \left( \frac{^H CO_2}{^L CO_2} \right)^s, \tag{6}$$

where $^H CO_2$ and $^L CO_2$ are the $CO_2$ concentrations that correspond to the two simulations in Equation 5. For surface air temperature, this process is equivalent to interpolating or extrapolating the straight lines in Figure 1(a) to identify the $CO_2$ that corresponds to $< T^p >$.

For CESM and GFDL the scaling is found by interpolation ($s < 1.0$) because there are simulations that are warmer than $< T^p >$. For those models where the scaling extrapolates beyond the model simulations (i.e. $s > 1.0$; COSMOS, HadCM3, IPSL, and NorESM), care must be taken due to the assumption of linearity. For HadCM3, IPSL, and COSMOS this assumption is probably well justified ($s$= 1.51, 1.37, and 1.05 respectively for the surface air temperature scaling), but for NorESM this assumption is probably poorly justified ($s$= 2.02).

For HadCM3, GFDL, IPSL, CESM, COSMOS, and NorESM, the inferred $CO_2$ for the surface air temperature scaling are 1030, 1050, 1080, 1130, 1140, and 2270 ppmv respectively. For CESM, COSMOS, and NorESM, these are consistent with the $CO_2$ proxy estimates of 1120-2520 ppmv (Section 3.4.1); the other models have a slightly lower inferred $CO_2$ than the proxies indicate. All these inferred $CO_2$ are below the concentration at which the respective models are known to crash (see Section 3.1). When the same method is applied to the EoMIP simulations, the inferred $CO_2$ are all higher than the proxy estimates (2640ppmv for HadCM3L, 3300ppmv for CCSM3_H, and 6210ppmv for CCSM_W). Figure 1 indicates that these relatively cool EoMIP simulations are related to a relatively low climate sensitivity for CCSM3_H and CCSM3_W and to a relatively low response to non-$CO_2$ forcing for HadCM3L and CCSM3_W.

The scaled SST anomalies, relative to the zonal mean of the preindustrial SST, for CESM, COSMOS, GFDL, HadCM3, IPSL, and NorESM, along with the proxy SST data from Hollis et al. (2019) (also relative to the zonal mean of the prein-dustrial), are shown in Figure 5. In general, the models agree reasonably well with the tropical and mid-latitude SST data, but there is a large model-data inconsistency in the southwest Pacific sites around New Zealand and south of Australia, where the modelled anomalies are colder than proxy estimates by 5–10 °C. See also Figure S7 in the Supplement for the modelled absolute SSTs and absolute SST proxy data.

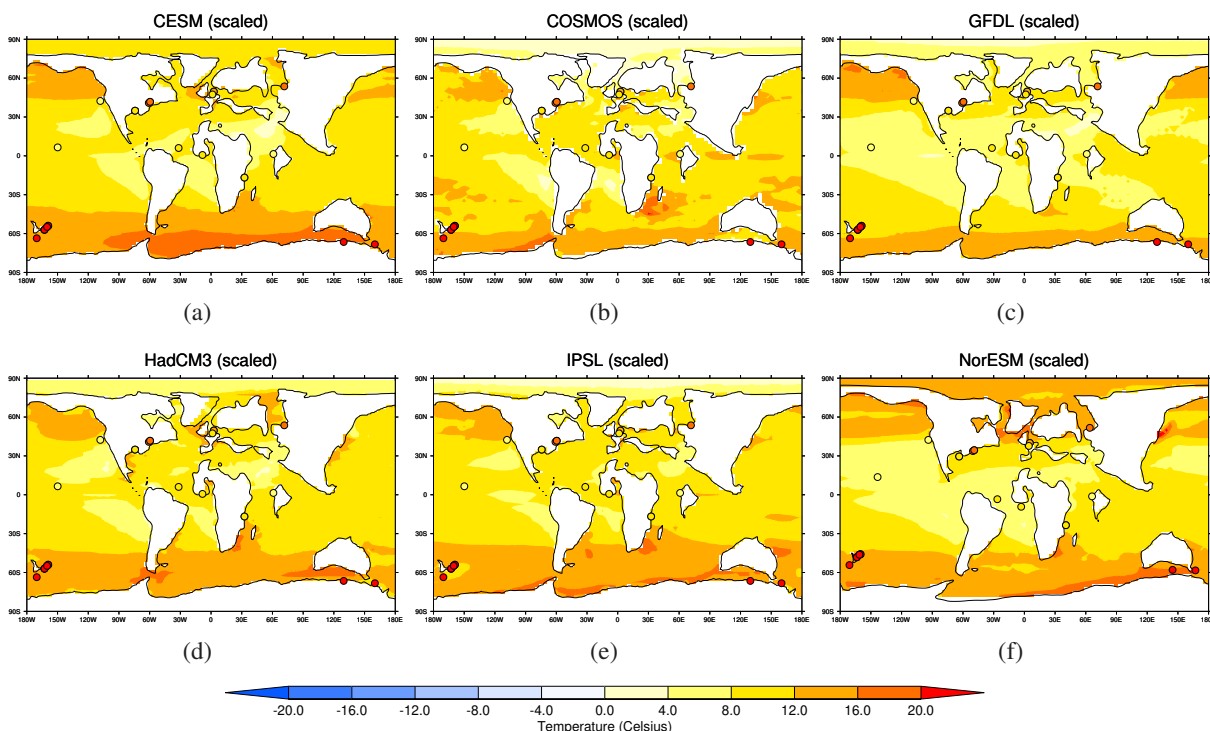

**Figure 5.** Modelled SST anomalies for the Eocene, relative to the zonal mean of the associated preindustrial simulation. The variable plotted is $SST_e^m - \overline{SST_e^m}$ in the notation of Lunt et al. (2012). The Eocene simulations have been scaled using a global tuning factor, as described in the text, so that they best fit the global mean SST data inferred from Inglis et al. (2020) (see Equation 5 and Equation 4). As such, only models that carried out simulations with more then one $CO_2$ concentration are shown. (a) CESM, (b) COSMOS, (c) GFDL, (d) HadCM3, (e) IPSL, (f) NorESM. Also shown are the proxy SST estimates from Hollis et al. (2019) for the EECO, excluding those sites that they identified as being affected by diagenesis.

The RMS skill-score of the scaled absolute simulations, relative to the SST proxies, $\sigma_s$ [°C], is 7.0 for NorESM, 9.6 for GFDL, 9.7 for CESM, 10.5 for HadCM3, 10.7 for IPSL, and 12.0 for COSMOS. Note that the NorESM score is not directly comparable to the others because the NorESM simulation, and the proxy data it is compared with, are on a paleomagnetic reference frame rather than a mantle reference frame (Figure 5(f)). For comparison, the GFDL skill-score is 7.3 when calculated on the paleomagnetic reference frame. Note that we calculate all RMS scores from a specific point-point comparison of models and data, not from zonal means.

To put these numbers in context, we also calculate the same skill score for some idealised temperature distributions (on the mantle reference frame), expressed as anomalies relative to the zonal mean of the preindustrial observations. This approach is similar to that used by Hargreaves et al. (2013) in the context of Quaternary model-data comparisons. Our idealised temperature distributions are (i) a constant value of zero (i.e. no change from the zonal mean of the preindustrial), (ii) a non-zero constant value, $C$, and (iii) a function $f(\phi) = A + B(1 - \cos 2\phi)$. For the constant value, $C$, we choose a value that is equal to an estimate

of global mean SST change from the proxies. This estimate of SST change is scaled from the proxy-based estimate of GSAT, $<T^p> = 27°C$, using the scaling in Equation 4, minus the preindustrial global mean SST. For the function $f(\phi)$, we choose $A$ and $B$ such that the global mean is equal to $C$, and the polar amplification metric, defined as the average SST equatorwards of $\pm 30°$ minus the average SST polewards of $\pm 60°$, is equal to our central estimate, i.e. 18°C minus the preindustrial value (see Section 3.4.1).

These idealised functions are shown in Figure 6(g-i) as zonal means, along with the scaled DeepMIP models (Figure 6(a-f)), all expressed as anomalies relative to the zonal mean of the preindustrial. The global anomaly of zero relative to the zonal mean of the preindustrial is associated with an RMS skill score $\sigma_s$=20.1, the global mean constant temperature anomaly, $C$, is associated with $\sigma_s$=11.6, and the $f(\phi)$ temperature profile is associated with $\sigma_s$=9.0 ($\sigma_s$=7.5 on the paleomagnetic reference frame). This means that all the models apart from COSMOS can be considered as having some skill in capturing the first-order patterns of SST change (because the skill score of those models is better than that of the global constant), but only NorESM has skill in capturing the second-order, more regional temperature patterns (because the skill score of the other models is worse than that of the $f(\phi)$ distribution when calculated on the appropriate reference frame). However, the performance of the scaled NorESM simulations should be viewed with some caution because of its relatively high scaling factor, $s$.

So far this analysis has focussed on SSTs, but we also compare with terrestrial near-surface air temperature data (SAT), even though it is in general less well constrained in age than SSTs, and as such likely represents a wider range of climate states. The absolute SAT model-data comparison for each DeepMIP simulation is shown in Figure S8 in the Supplement. For those models that carried out more than one $CO_2$ simulation (CESM, COSMOS, GFDL, HadCM3, IPSL, and NorESM), Figure S9 and S10 in the Supplement show the SATs from the scaled models in comparison with terrestrial proxy data.

These figures show that both models and SAT terrestrial proxies show a similar amount of polar amplification. In particular, the southwest Pacific site SATs are better simulated in the models than the SSTs; the RMSE error decreases in the southwest Pacific by 30% on average for the SATs compared with the SSTs across the ensemble. This implies that there may be an inconsistency between marine and terrestrial temperatures in either the proxies or models in this region. This discrepancy could be related to a potential summer bias in mid and high latitude SST proxies (Hollis et al., 2012; Davies et al., 2019). An alternative hypothesis is that the discrepancy is related to Red Sea-like features of GDGT distributions in high SST samples from the southwest Pacific and Wilkes Land, that appear to amplify proxy SSTs where $iso$GDGT$_{RS}$ > 30 (Inglis et al., 2015), an idea supported by recent work in the context of the Cretaceous (Steinig et al., 2020). However, the discrepancy may also be caused by physical processes that are not captured by any of the models.

## 4 Conclusions

We have presented an ensemble of model simulations of the Eocene, carried out in the framework of DeepMIP. Focus has been on documenting the annual mean modelled temperatures, exploring the reasons for the different responses in the models, and comparing with proxy data. Compared with previous model simulations, the results show reduced spread across the ensemble (excepting the INMCM model), and greater climate sensitivity on average. The contribution to Eocene warmth from non-$CO_2$

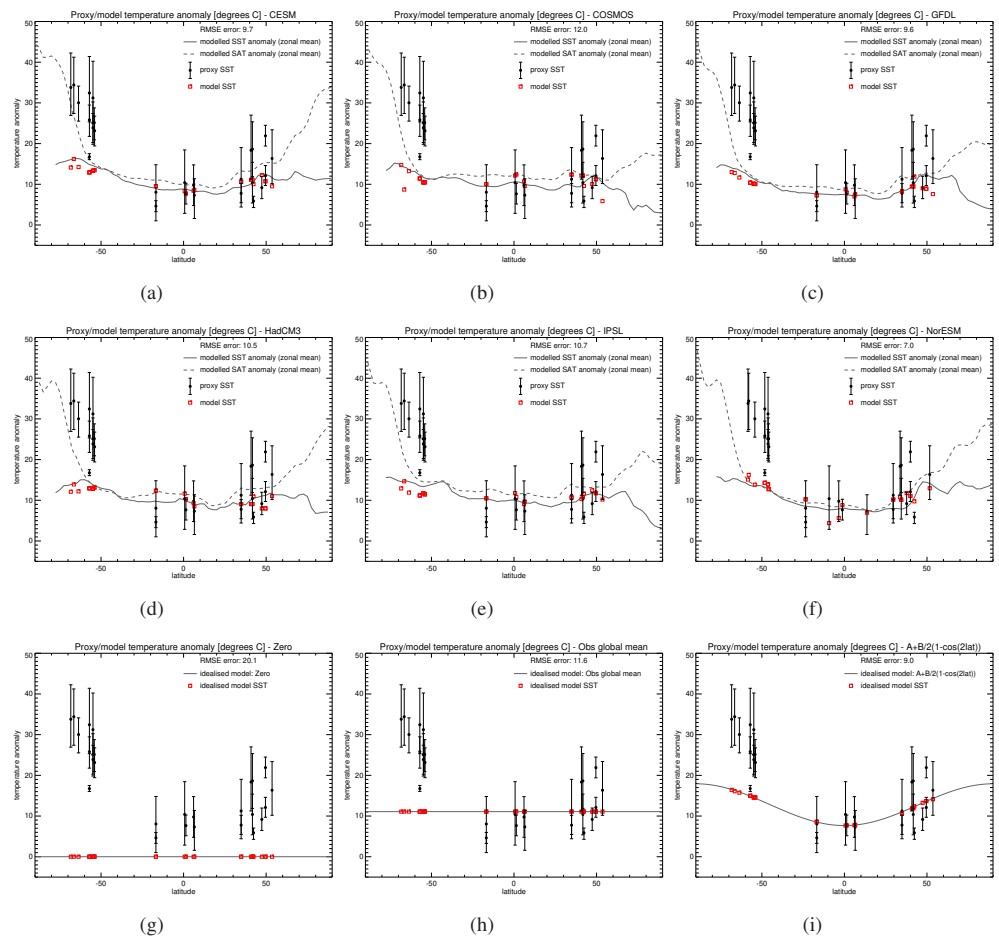

**Figure 6.** (a,b,c,d) Zonal mean SST (solid lines) and near-surface air temperature (dashed line) anomalies, relative to the zonal mean of the associated preindustrial simulation, for the scaled version of the (a) CESM, (b) COSMOS, (c) GFDL, (d) HadCM3, (e) IPSL, and (f) NorESM models. Also shown are the EECO SSTs and error bars from Hollis et al. (2019), also expressed as a difference relative to the zonal mean of the preindustrial. (g,h,i) As (a,b,c,d,e,f) but instead of a model we show idealised temperature distributions of (g) 0, (h) $C$ and (i) $A + B(1 - \cos 2\phi)$. All plots also show the proxy SST estimates from Hollis et al. (2019) for the EECO, excluding those sites that they identified as being affected by diagenesis (black circles with uncertainty bars). Also shown are the modelled/idealised SSTs at the specific location of the proxies (red squares).

boundary conditions (paleogeography and vegetation and aerosols) is between 3.1°C (HadCM3) and 5.1°C (CESM). The reasons for the model spread is explored using an energy balance framework. This indicates that the difference between the models with the greatest and least warming in the Eocene at $\times 3$ CO$_2$ is due primarily to differences in the surface albedo response, and that the difference between the models with the greatest and least polar amplification in the Eocene is due primarily to differences in the albedo and emissivity response. Across the model ensemble, the global mean warming in the

Eocene compared with preindustrial arises mostly from changes in emissivity due to the elevated CO$_2$ and associated water

vapour and longwave cloud feedbacks, whereas in terms of the meridional temperature gradient, the reduction in the Eocene is primarily due to emissivity and albedo changes due to the non-$CO_2$ boundary conditions (i.e. removal of the Antarctic ice sheet and changes in vegetation). Due to the limited range of prescribed $CO_2$ in the model simulations, coupled with uncertainties in proxy reconstructed $CO_2$, we interpolate and extrapolate between simulations at multiple $CO_2$ concentrations to infer the concentration that gives the best fit to previous estimates of the global mean temperature, and then compare the model inferred temperatures to the proxy SSTs on a point-by-point basis. This shows that CESM, GFDL, HadCM3, IPSL, and NorESM all have "skill" in representing the first-order patterns in the SST proxies in that they agree better with the proxies than a tuned global constant warming. However, they do not reproduce the exceptional warmth in the southwest Pacific proxy SSTs, although the modelled and proxy SATs are in better agreement than SSTs in this region, pointing to a possible inconsistency between the marine and terrestrial paleo temperatures in either the models or the proxies. Despite the regional limitations in the SST model-proxy consistency, the scaled CESM, COSMOS, and NorESM models all simulate a best-fit global mean temperature at $CO_2$ concentrations that are consistent with the $CO_2$ proxies, without prescribing changes to model parameters such as those related to clouds. Furthermore, CESM, GFDL, and NorESM are all consistent with the global mean temperature, meridional temperature gradient, and $CO_2$ proxies. Tighter constraints from proxies on both $CO_2$ and temperature would allow better discrimination of the DeepMIP models and model simulations. It is worth noting that CESM and GFDL both implemented modified aerosols in their Eocene simulations (see Section 2.2); the importance of this remains a topic for further investigation. Other future work in the framework of DeepMIP will explore the model simulations and model-data comparisons in more detail, in particular the response of clouds, the hydrological cycle, and ocean circulation.

*Data availability.* The model results, in terms of annual mean near-surface air temperature, SST, and radiative fluxes, for the Eocene and preindustrial control simulations, are available in the Supplement as netcdf files. These are derived from files in version 1.0 of the DeepMIP model database, by interpolating to a common grid ($3.75°$ longitude $\times 2.5°$ latitude), using cdo operators. Bilinear interpolation is used for the near-surface air temperature data and nearest-neighbour interpolation is used for the SST data. Access to the full DeepMIP model database can be requested from the corresponding author. The proxy database used in this study is identical to that used in Inglis et al. (2020), and is available from the Supplement of that study. This contains the same data as in Hollis et al. (2019), i.e. version 0.1 of the DeepMIP proxy database.

*Author contributions.* All authors contributed to the writing of the paper. DJL carried out the analysis. The model simulations were carried out by JZ (CESM), IN (COSMOS), DKH (GFDL), FB and SS (HadCM3), PM and EV (INMCM), JBL,PS and YD (IPSL), WLC (MIROC), and ZZ (NorESM). CJH and TDJ led the compilation of the DeepMIP database. DJL, MH, and BLOB coordinated the study.

*Competing interests.* The authors declare no competing interests.

*Acknowledgements.* DL, SB, and FB thank NERC grant SWEET (NE/P01903X/1). DL thanks NERC grant DeepMIP (NE/N006828/1) and ERC grant "The greenhouse earth system" (T-GRES, project reference 340923, awarded to Rich Pancost). CP and JT acknowledge the Heising-Simons Foundation Grant #2016-015. JZ and CP thank J. Kiehl, C. Shields, and M. Rothstein for providing the CESM code and boundary/initial condition files for CESM simulations. WLC and AAO acknowledge funding from JSPS KAKENHI grant 17H06104 and MEXT KAKENHI grant 17H06323, and JAMSTEC for use of the Earth Simulator. DH and AdB were supported by the Swedish Research Council Project 2016-03912 and FORMAS Project 2018-01621. Their numerical simulations were performed using resources provided by the Swedish National Infrastructure for Computing (SNIC) at NSC, Linköping. PS, JBL and YD were granted access to the HPC resources of TGCC under the allocation 2019-A0050102212 made by GENCI. The HadCM3 simulations were carried out using the computational facilities of the Advanced Computing Research Centre, University of Bristol - http://www.bristol.ac.uk/acrc/. GNI acknowledges a Royal Society Dorothy Hodgkin Fellowship. MH was funded by the US National Science Foundation (NSF) grants ATM-0902780 and OCE-0902882. BLO-B acknowledges the CESM project, which is supported primarily by the National Science Foundation (NSF). This material is based upon work supported by the National Center for Atmospheric Research (NCAR), which is a major facility sponsored by the NSF under Cooperative Agreement No. 1852977. Computing and data storage resources, including the Cheyenne supercomputer (doi:10.5065/D6RX99HX), were provided by the Computational and Information Systems Laboratory (CISL) at NCAR. TDJ was supported by NERC grant NE/P013112/1. PM was supported by the state assignment project 0148-2019-0009.

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
