# Peer review of "DeepMIP: Model intercomparison of early Eocene climatic optimum (EECO) large-scale climate features and comparison with proxy data"

_Climate of the Past, 2019_

## Referee Comment (RC1) · Anonymous Referee #1 · 19 Feb 2020

The authors present new results from EECO simulations from seven different climate models. The simulations have been carried out in the framework of the DeepMIP project. The study considers large-scale climatic features (global surface temperature and meridional temperature gradients in relation to CO2) and performs some basic model-data comparison. My main criticism is that this article is largely descriptive. The authors state that "the aim here is not to fully understand the whole ensemble from a mechanistic viewpoint, but to document the large-scale features". Though I understand that this manuscript is probably only a prelude to further DeepMIP studies, this study

should be improved by adding at least a minimum of analysis that provides some insight into mechanisms and processes. I therefore suggest to add some analyzes regarding the polar amplification in both hemispheres, which is obviously very different in the different models. Why do CESM and GFDL show much stronger polar amplification than COSMOS, HadCM3B and IPSL? What is the role of sea ice loss and at which CO2 concentrations does the sea ice disappear in the different models? What is the role of increasing water vapor and clouds, what is the role of a changing polar temperature profile, etc.? The authors should provide at least some basic analysis (e.g., see Dai et al., 2019, Nature Comm., https://doi.org/10.1038/s41467-018-07954-9) since the polar amplification is a key aspect in these EECO simulations. Such analysis would not only substantially improve this manuscript, but also our understanding of Eocene and future climate.

Minor points: COSMOS was initialized with a homogeneous ocean and integrated for only 1000 years. This is probably too short and trends are still likely to affect the results. How large are these trends? Please provide some numbers or a supplementary figure showing e.g. temperature and salinity time series, or – even better – continue the integration until equilibrium is reached.

Figure 1 is a mess. Please revise the figure such that model names become readable or – if this is not possible – remove the model names from the diagram and use colors and symbols instead. As it is now, this figure is not publishable.

Figure 3: "GFD" should be "GFDL".

Line 421: I guess HCO2 and LCO2 are not CO2 concentrations but multiples of the pre-industrial concentration.

---

## Referee Comment (RC2) · Anonymous Referee #2 · 26 Mar 2020

Review of the manuscript by Lunt et al. entitled "DeepMIP: Model intercomparison of early Eocene climatic optimum (EECO) large-scale climate features and comparison with proxy data"

The manuscript provides a nice overview of state-of-the-art climate modelling of the EECO and comparison with proxy reconstructions. It is well written and structured. Below I will list my main concerns that I think should be remedied before publication,

together with a number of minor and technical comments.

Main comments:

The interval that is given for reconstructed $CO_2$ concentrations is 900-2500ppm (95To circumvent this issue, a methodology is applied to estimate the simulated temperature response at $CO_2$ values that have not actually been simulated, an interesting idea. However, extrapolating too far outside of the range of simulated $CO_2$ concentrations is difficult because of the possible non-linear relationships in the models, as clearly explained in the manuscript. For this reason, the authors limit the amount of extrapolation, introducing quite some ambiguity. In turn this leads to the situation that even using this extrapolation method, most models cannot estimate the 'best' $CO_2$ value, only a 'minimum estimate' as the authors call it. However, in the remainder of the results section these simulations are presented as 'tuned to best fit the SST proxy data', which I find misleading. Moreover, using these extrapolations as if they where actual fully coupled climate simulations and compare them with site specific SST records, as is done for the southwest Pacific, New Zealand and Australia, really seems a bridge too far for me. This seems to be acknowledge by the authors in the subsequent analysis that they present on lines 431-440, however, still the regional model-data 'mismatches' are presented and even listed in the abstract and conclusion section. Please clarify the validity of this approach and the 'weight' that seems to be given to these regional model-data comparisons.

Minor comments:

What is the reason that simulations with higher $CO_2$ levels are often not performed? For some models it is mentioned that they become unstable for such high $CO_2$ levels and if this is a general problem, it seems that this is worth mentioning.

Lines 20-21: 'equivalent models' is a somewhat vague term that could hide the fact that only 1 out of 7 models is used in CMIP6 and only 3 out of 7 in CMIP5, the other four models are CMIP3. This should me mentioned more clearly.

Lines 70-72: Similar to the comment above, you mention that many of the current-generation models include improved treatment of cloud processes, however, most of the models that are used are not current-generation models. Please clarify.

Lines 452-255: These are again the minimum $CO_2$ levels estimated from the models, not the 'best' ones? Please clarify.

Line 461: Uncertainty in the reconstructed $CO_2$ concentrations is only one of the reasons to apply interpolation and extrapolation of the model results, just as important or perhaps even the most important reason is that only four simulations with appropriate $CO_2$ levels are available (from a total of only three different models out of seven).

Lines 463-467: CESM and GFDL are also the only two models that did high $CO_2$ simulations (6x and 9x), with 6x being close to the middle of the estimate $CO_2$ range of 900-2500ppm. The only other simulation that is within this range (NorESM with 4x $CO_2$) is at the lower end of this range. So it seems that this is a must simpler explanation for why CESM and GFDL are the 'best' models, without the need for a statement about the implemented modified aerosol schemes.

Technical comments:

Line 224: "step-wise using", word missing?

Line 346: correct "abd"

Line 356: the word 'also' seems strange here since the previous sentence discusses differences between models, not similarities.

Figure 1: The labels cannot be read this way, another way of presenting that information must be found. The CCSM3 data is difficult to read, please update.

Data availability: I was not able to find any netcdf files in the supplement, but perhaps that is still to come?

---

## Author Comment (AC1) · 26 Aug 2020

Please see the response to Reviewer 1, which is a combined response.

---

## Author Comment (AC2) · 26 Aug 2020

please see attached for a response to both reviewers.

Please also note the supplement to this comment:
https://cp.copernicus.org/preprints/cp-2019-149/cp-2019-149-AC2-supplement.pdf

---

## Author Response (AR1)

We thank the two reviewers for their very helpful comments on the manuscript, which have greatly improved the paper. Here we provide a point-by-point response to those comments. The line numbers **in blue** refer to the marked-up ('latexdiff') version of the manuscript, which indicates how this revised manuscript compares to the originally submitted version (note that in the model description Section 2.2 the changes appear substantial, but are largely just a re-ordering of the models so that they now appear alphabetically).

**Reviewer 1:**

My main criticism is that this article is largely descriptive. The authors state that "the aim here is not to fully understand the whole ensemble from a mechanistic viewpoint, but to document the large-scale features". Though I understand that this manuscript is probably only a prelude to further DeepMIP studies, this study should be improved by adding at least a minimum of analysis that provides some insight into mechanisms and processes. I therefore suggest to add some analyzes regarding the polar amplification in both hemispheres, which is obviously very different in the different models. Why do CESM and GFDL show much stronger polar amplification than COSMOS, HadCM3B and IPSL? What is the role of sea ice loss and at which CO2 concentrations does the sea ice disappear in the different models? What is the role of increasing water vapor and clouds, what is the role of a changing polar temperature profile, etc.? The authors should provide at least some basic analysis (e.g., see Dai et al., 2019, Nature Comm., https://doi.org/10.1038/s41467-018-07954-9) since the polar amplification is a key aspect in these EECO simulations. Such analysis would not only substantially improve this manuscript, but also our understanding of Eocene and future climate.

**We agree that, even though this is an overview paper with follow-up papers in preparation, it is good to include a more detailed analyses for the reasons behind the different responses in the different models. As such, we have added a new section that explores this in detail (Section 3.3). In particular, we firstly present and discuss the spatial changes in surface albedo, planetary albedo, and emissivity (Figures S2 to S4 in Supp info). We then apply the energy-balance approach of Heinemann et al (2009) to partition the differences in zonal-mean surface temperature in the simulations into components related to surface albedo, TOA albedo, emissivity, and heat transport. We further partition these changes into components due to non-CO$_2$ boundary conditions (paleogeography, vegetation, ice, and aerosols) and due to CO$_2$. This analysis is presented in Figure 4 (and Figures S5 and S6 in Supp Info), and summarised, in terms of contributions to the global mean and to polar amplification, in Table 2. We think that this provides a comprehensive and quantitative understanding for the differences in the model responses, on a global scale.**
**See Section 3.3; lines 535-638.**

COSMOS was initialized with a homogeneous ocean and integrated for only 1000 years. This is probably too short and trends are still likely to affect the results. How large are these trends? Please provide some numbers or a supplementary figure showing e.g. temperature and salinity time series, or – even better – continue the integration until equilibrium is reached.
**The 3x and preindustrial simulations of COSMOS have now been integrated much further, until year 9500. In addition, two simulations were branched off from this simulation (with 1x and 4x pre-industrial CO2 levels) from year 1000. Both simulations were further run until year 9500. More details can be found in the revised version in the COSMOS model description subsection. Furthermore, we now add a short section (Section 3.1) on model equilibration and spinup in all the simulations, referring back to the original criteria for equilibration set out in the experimental design paper (Lunt et al, 2017).**
**See Section 2.2.2; lines 219-222 and Section 3.1; lines 428-451.**

Figure 1 is a mess. Please revise the figure such that model names become readable or – if this is not possible – remove the model names from the diagram and use colors and symbols instead. As it is now, this figure is not publishable.
**We agree, and have substantially revised Figure 1. We have removed the DeepMIP model names and replaced these with a legend, and made the symbols larger. We now have colours corresponding to models rather than to CO2 concentrations. We also modified the axes and axes labels as appropriate.**
**See new Figure 1.**

Figure 3: "GFD" should be "GFDL".
**Fixed.**
**See new Figure 3.**

Line 421: I guess HCO2 and LCO2 are not CO2 concentrations but multiples of the pre-industrial concentration

**Removed the 280 in the equation so that $^{H}CO_2$ etc. are absolute $CO_2$ values.**
See Equation 6, line 741.

**Reviewer 2:**

The interval that is given for reconstructed CO2 concentrations is 900-2500ppmv. To circumvent this issue, a methodology is applied to estimate the simulated temperature response at CO2 values that have not actually been simulated, an interesting idea. However, extrapolating too far outside of the range of simulated CO2 concentrations is difficult because of the possible non-linear relationships in the models, as clearly explained in the manuscript. For this reason, the authors limit the amount of extrapolation, introducing quite some ambiguity. In turn this leads to the situation that even using this extrapolation method, most models cannot estimate the 'best' CO2 value, only a 'minimum estimate' as the authors call it. However, in the remainder of the results section these simulations are presented as 'tuned to best fit the SST proxy data', which I find misleading. Moreover, using these extrapolations as if they were actual fully coupled climate simulations and compare them with site specific SST records, as is done for the southwest Pacific, New Zealand and Australia, really seems a bridge too far for me. This seems to be acknowledge by the authors in the subsequent analysis that they present on lines 431-440, however, still the regional model-data 'mismatches' are presented and even listed in the abstract and conclusion section. Please clarify the validity of this approach and the 'weight' that seems to be given to these regional model-data comparisons.

**We agree that the scaling proposed was problematic. As such, we now scale to the global mean temperature presented in Inglis et al (2020), instead of by minimising the RMSE relative to the individual proxy data. This has the effect that the scaling factor is now substantially smaller for all models, meaning that we are either interpolating rather than extrapolating, or at least extrapolating less far. The scaling factor is smaller than previously because the global mean is less influenced by the anomalously warm sites in the SW Pacific. This also reduces the inherent circularity of tuning to the individual data points and then comparing to that same dataset. Furthermore, we explicitly present the scaling factors, allowing the reader to assess the implication of this process. Finally, due to the reduced scaling we no longer "cap" the process, and so the results are more straightforward to interpret.**
See Section 3.4.3; lines 718-781.

What is the reason that simulations with higher CO2 levels are often not performed? For some models it is mentioned that they become unstable for such high CO2 levels and if this is a general problem, it seems that this is worth mentioning.

**With relevance to the previous comment, for those simulations that were scaled in the model-data comparison process, we give the $CO_2$ concentration at which they developed instabilities and blew up. For all models, the inferred $CO_2$ was below the $CO_2$ concentration at which the model is known to develop instabilities.**
See Section 3.1; lines 449-451 and Section 3.4.3; lines 748-754.

Lines 20-21: 'equivalent models' is a somewhat vague term that could hide the fact that only 1 out of 7 models is used in CMIP6 and only 3 out of 7 in CMIP5, the other four models are CMIP3. This should me mentioned more clearly.

**Changed to "Paleoclimate model-data comparisons allow us to assess confidence in the results from model sensitivity studies that explore the mechanisms that drove past climate change, and allow us to assess confidence in the future climate predictions from these models"**
See lines 26-29.

Lines 70-72: Similar to the comment above, you mention that many of the current generation models include improved treatment of cloud processes, however, most of the models that are used are not current-generation models. Please clarify.

**Changed to "Furthermore, some models are available for deep-time paleoclimate simulations that are more advanced than those used in the Lunt et al (2012) study; for example CESM1.2 includes a more advanced cloud microphysics scheme compared with CCSM3, HadCM3 has a higher ocean resolution than HadCM3L, and INMCM and NorESM are CMIP6-class models and therefore can be considered state-of-the-art."**
See lines 80-84.

Lines 452-255: These are again the minimum CO2 levels estimated from the models, not the 'best' ones? Please clarify.

**With our revised scaling, the estimates of $CO_2$ are now "best" estimates, because we no longer cap the scaling (and see response to previous comment).**

Line 461: Uncertainty in the reconstructed CO2 concentrations is only one of the reasons to apply interpolation and extrapolation of the model results, just as important or perhaps even the most important reason is that only four simulations with appropriate CO2 levels are available (from a total of only three different models out of seven).
**"The limited range of $CO_2$ concentrations explored by some models, coupled with…."**
**See line 719.**

Lines 463-467: CESM and GFDL are also the only two models that did high CO2 simulations (6x and 9x), with 6x being close to the middle of the estimate CO2 range of 900-2500ppm. The only other simulation that is within this range (NorESM with 4x CO2) is at the lower end of this range. So it seems that this is a must simpler explanation for why CESM and GFDL are the 'best' models, without the need for a statement about the implemented modified aerosol schemes.
**Because of the new scaling methodology, this comment no longer applies.**

Line 224: "step-wise using", word missing?
**Changed to "adjustment"**

Line 346: correct "abd"
**Done.**

Line 356: the word 'also' seems strange here since the previous sentence discusses differences between models, not similarities.
**Removed "also".**

Figure 1: The labels cannot be read this way, another way of presenting that information must be found. The CCSM3 data is difficult to read, please update.
**We have substantially revised Figure 1. We have removed the DeepMIP model names and replaced these with a legend, and made the symbols larger. We now have colours corresponding to models rather than to CO2 concentrations. We also modified the axes and axes labels as appropriate.**
**See new Figure 1.**

Data availability: I was not able to find any netcdf files in the supplement, but perhaps that is still to come?
**The netcdf files are now included in the Supplement.**

**Other changes:**

(1) We changed our definition of our meridional temperature gradient metric in the models. Previously it was the mean of the zonal means in the latitudinal range, now it is the mean of all non-ocean gridcells in the latitudinal range (Figure 1b, y axis, and associated Caption).

(2) We modified the range of proxy data in Figure 1 to reflect a more recent work. In particular global-mean near surface air temperature is given according to Inglis et al (in press, Climate of the Past), and for meridional temperature gradient we incorporate the studies of Evans et al (2018) and Zhu et al (2019) in addition to Cramwinckel et al (2018). Furthermore, in this Figure, one of the FAMOUS simulations was mis-plotted and this has been corrected. We added an associated Section 3.4.1. that describes the proxy datasets.

(3) We have updated some of the model output. In particular, this includes longer simulations for HadCM3B and COSMOS, and the addition of a new model, INMCM.

(4) We made a few editorial changes, e.g. ensuring the naming of models, and the order in which they are presented and discussed, is consistent throughout the manuscript and figures.

(5) Because of the above additions and changes in response to the reviewer comments, the abstract and conclusions have been modified where appropriate.

[revised manuscript text omitted]

---

## Author Response (AR2)

We thank the reviewer for their comments on the manuscript. Here we provide a point-by-point response. The line numbers **in blue** refer to the marked-up version of the manuscript.

**Reviewer 1:**

Line 350: Explain the difference between top-of-the-model and top-of-the-atmosphere and how this affects the TOA inbalance.
**Agreed – added:**
**"because there is some atmosphere above the top of the model that can interact with incoming or outgoing radiation (i.e. the model top is not at 0 mbar)."**
**See Section 3.1; lines 352-353.**

Line 357: Provide some reasons (at least speculations) for why the models crashed under high CO2.
**Agreed - added:**
**"These crashes have not been explored in detail, but could be due to feedbacks becoming more positive as temperature increases (for example associated with an increase in height of the tropopause; Meraner et al., 2013), to such an extent that positive feedbacks overcome the negative Planck feedback (Bloch-Johnson et al., 2015), at which point a "runaway" phase is entered and the temperature begins to increase rapidly. This can then cause violation of the CFL criterion due to high wind speeds associated with the generation of large pressure and/or temperature gradients, causing the model to crash."**
**See Section 3.1; lines 361-366.**

line 609: "assumption is less well justified". For s=2.02 I would say, this assumption is poorly justified. Please rephrase.
**Agreed - rephrased to:**
**"assumption is probably poorly justified".**
**See Section 3.4.3; line 616.**

**Other changes:**

(1) Added to Data Availability Statement that radiative fluxes are available in the Supplement.
(2) Modified captions for Figure 6, S7, and S8 slightly.
(3) Added: "Tighter constraints from proxies on both $CO_2$ and temperature would allow better discrimination of the DeepMIP models and model simulations."
(4) Added/modified some references to the model description section and Table 1.
(5) Mentioned that MIROC also carried out simulations at x1 and x2, although they are not discussed in this manuscript.
(6) A small number of minor grammatical typos were fixed.

[revised manuscript text omitted]

---

## Author Response (AR3)

We thank the Editor for handling the manuscript so thoroughly and efficiently.

At this final stage we made a small number of very minor non-scientific changes, as listed below:

(1) We expanded some of the author affiliations on the title page.
(2) We expanded some of the Acknowledgments.
(3) We updated the "Inglis et al, Climate of the Past Discussions, 2020" reference to "Inglis et al, Climate of the Past, 2020"
(4) We added some references to the bibliography which appeared in the text but not the reference list.
(5) We corrected some minor typos in the INMCM model description section.
(6) We added Evgeny Volodin as a co-author and to the Author Contributions section (he contributed to the INMCM model simulations). We checked and all other co-authors approve of this change.
(7) We added "Open square shows modern observations." to the caption of Figure 1.
(8) We made the sign of fluxes in the netcdf files in the COSMOS model in Supp Info consistent with the other models, and removed the text in Supp Info that commented on these being inconsistent.

See the changes above in the "latexdiff" version of the paper that follows.

Given that these changes are so minor, I ticked the box saying that "I confirm that the submitted manuscript is in the same form as that accepted by the Editor".

Yours,

Dan Lunt

[revised manuscript text omitted]